# Correlative cryo-electron microscopy reveals the structure of TNTs in neuronal cells

Anna Sartori-Rupp[1], Diégo Cordero Cervantes[2], Anna Pepe[2], Karine Gousset [2,3], Elise Delage[2], Simon Corroyer-Dulmont[1], Christine Schmitt[1], Jacomina Krijnse-Locker[1] & Chiara Zurzolo[2]

The orchestration of intercellular communication is essential for multicellular organisms. One mechanism by which cells communicate is through long, actin-rich membranous protrusions called tunneling nanotubes (TNTs), which allow the intercellular transport of various cargoes, between the cytoplasm of distant cells in vitro and in vivo. With most studies failing to establish their structural identity and examine whether they are truly open-ended organelles, there is a need to study the anatomy of TNTs at the nanometer resolution. Here, we use correlative FIB-SEM, light- and cryo-electron microscopy approaches to elucidate the structural organization of neuronal TNTs. Our data indicate that they are composed of a bundle of open-ended individual tunneling nanotubes (iTNTs) that are held together by threads labeled with anti-N-Cadherin antibodies. iTNTs are filled with parallel actin bundles on which different membrane-bound compartments and mitochondria appear to transfer. These results provide evidence that neuronal TNTs have distinct structural features compared to other cell protrusions.

[1] Institut Pasteur, Unit of Technology and Service Ultra-structural Bio-Imaging, 28 rue du Docteur Roux, 75015 Paris, France. [2] Institut Pasteur, Membrane Traffic and Pathogenesis, 28 rue du Docteur Roux, 75015 Paris, France. [3] Present address: Department of Biology, College of Science and Math, California State University, Fresno, 2555 East San Ramon Avenue M/S SB73, Fresno, CA 93740-8034, USA. These authors contributed equally: Anna Sartori-Rupp, Diégo Cordero Cervantes, Anna Pepe. Correspondence and requests for materials should be addressed to C.Z. (email: chiara.zurzolo@pasteur.fr)

Tunneling nanotubes (TNTs) have been defined as long, thin, non-adherent membranous structures that form contiguous cytoplasmic bridges between cells over long and short distances ranging from several hundred nm up to 100 μm[1–4]. Over the last decade, scientific research has effectively improved our understanding of these structures and underscored their role in cell-to-cell communication, facilitating the bi- and unidirectional transfer of compounds between cells, including: organelles, pathogens, ions, genetic material, and misfolded proteins[5]. Altogether, in vitro and in vivo evidence has shown that TNTs can be involved in many different processes such as stem cell differentiation, tissue regeneration, neurodegenerative diseases, immune response, and cancer[2,6–10].

Although these in vitro and in vivo studies have been informative, the structural complexity of TNTs remains largely unknown. One of the major issues in this field is that many types of TNT-like connections have been described using mainly low-resolution imaging methods such as fluorescence microscopy (FM). As a result, information regarding their structural identity and if or how they differ among each other and with other cellular protrusions such as filopodia, is still lacking. As a result, TNTs have been regarded with skepticism by one part of the scientific community[5,11].

Two outstanding questions are whether these protrusions are different from other previously studied cellular processes such as filopodia[12] and whether their function in allowing the exchange of cargos between distant cells is due to direct communication between the cytoplasm of distant cells or to a classic exo-endocytosis process or a trogocytosis event[13,14].

Addressing these questions has been difficult due to considerable technical challenges in preserving the ultrastructure of TNTs for electron microscopy (EM) studies. To date, only a handful of articles have examined the ultrastructure of TNTs using scanning and transmission EM (SEM and TEM, respectively)[1,15–18], and no correlative studies have been performed to ensure that the structures identified by TEM/SEM represent the functional units observed by FM.

Although very similar by FM, TNT formation appears to be oppositely regulated by the same actin modifiers that act on filopodia[19]. Furthermore, filopodia have not been shown to allow cargo transfer[12,20,21]. Thus, we hypothesize that TNTs are different organelles from filopodia and might display structural differences in morphology and actin architecture.

In order to compare the ultrastructure and actin architecture of TNTs and filopodia at the nanometer resolution we employed a combination of live imaging, correlative light- and cryo-electron tomography (ET) approaches on TNTs of two different neuronal cell models, (mouse cathecholaminergic CAD cells and human neuroblastoma SH-SY5Y cells)[19,22–25]. We found that single TNTs observed by FM are in most cases made up of a bundle of individual TNTs (iTNTs), each surrounded by a plasma membrane and connected to each other by bridging threads containing N-Cadherin. Each iTNTs appeared filled by one highly organized parallel actin bundle on which vesicles, mitochondria, and other membranous compartments appear to be traveling. Finally, by using correlative focused-ion beam SEM (FIB-SEM) we show that TNTs can be open on both ends, thus challenging the dogma of a cell as an individual unit[26].

Collectively, our data demonstrates that TNTs connecting neuronal cells are different cellular structures from other membrane protrusions, identifies the structural features that characterize this specific organelle, and supports their role in allowing direct communication between the cytosol of distant cells and cargo transfer.

## Results

**Ultrastructural analysis of TNTs reveals individual TNTs.** TNTs connections reported between different cells in culture appear to be very heterogeneous at the resolution of FM[22], therefore their structural identity is unclear. The elective method to study the ultrastructure of organelles and cellular features is correlative light- and electron microscopy. However initial attempts to analyze TNTs in neuronal CAD cells (Supplementary Fig. 1a)[19,22] by most routinely used correlative SEM showed that most TNTs broke during sample preparation (Supplementary Fig. 1b–e, g, h). Only few, thick, more robust TNTs were preserved (Supplementary Fig. 1f)[1,15,16]. Intriguingly, some of the broken structures appeared to be comprised of multiple smaller tubes (yellow arrowheads, Supplementary Fig. 1e, h).

To better preserve these fragile structures, we established correlative assays by combining cryo-FM with cryo-electron microscopy (cryo-EM) and cryo-electron tomography (cryo-ET) (see workflow in Fig. 1a). FM was used to screen for long (≥10 μm) and direct cell-to-cell connections labeled with wheat-germ agglutinin (WGA) hovering over the substrate, a phenotypic criterium used to identify TNTs[1,2,19,23,27]. Two correlative light and electron microscopy (CLEM) approaches were employed to image cells grown on EM grids: i-CLEM and ii-CLEM (see material and methods, Fig. 1a). By i-CLEM, we found that most TNTs which appeared as one tube under FM and low-magnification cryo-TEM (Fig. 1b–c) were comprised of a bundle of individual tunneling nanotubes (iTNTs) (Fig. 1d–h), each delimited by its own plasma membrane, as suggested by our SEM images (Supplementary Fig. 1d, e). iTNTs contained actin filaments and ran mostly parallel to and occasionally braided over each other (Fig. 1e, g). We also observed that their tips originated from opposite directions, suggesting that they were in the process of extending to other cells or retracting from opposite cells (red stars, Fig. 1f, h). Single thicker TNTs (600–900 nm in diameter), were rarely observed (Supplementary Fig. 1i–l).

TNTs have previously been reported to transfer vesicles, organelles, and various cargoes between cells by FM and live imaging;[2,4,5,22] however, whether these cargoes were transported through the TNT lumen or by surfing on the limiting membrane, as shown for viruses on filopodial bridges[28], could not be shown due to the resolution limitations of FM. By cryo-EM, we were able to observe vesicular membrane compartments located inside and between iTNTs. Inside TNTs, vesicles were heterogeneous in size and shape, spanning from single-membrane spherical vesicles (yellow arrowheads, Fig. 1g, h and Supplementary Fig. 2a–e, 2g) with an average diameter of 109 nm (SD = 25 nm) (Supplementary Fig. 2i), to multi-vesicular compartments (blue arrowheads, Supplementary Fig. 2f, h). This observation supports our previous FM-based studies in CAD cells showing TNT-mediated transfer of DiD-labeled vesicles, lysosomes, and aggregated proteins between cells[7,22,27], and indicates that transfer occurs through the iTNT lumen. Intriguingly, we also observed vesicles inside iTNTs with tips having opposite orientations (green arrowheads indicate vesicles, and red stars show the tips of tubes, Fig. 1h). This finding suggests that transport is unidirectional within each iTNT having an antiparallel orientation (see discussion). Vesicles found externally were enclosed by a double membrane (turquoise arrowheads, Fig. 1g and Supplementary Fig. 2e, h), suggesting that they resulted from single-membrane vesicles budding off the plasma membrane of an iTNT. Several vesicles coupled together surrounded by an outer membrane were also observed outside iTNTs (pink arrowheads, Fig. 1h and Supplementary Fig. 2b, c), which might be explained by a membrane pinch-off or a break of the iTNT containing them.

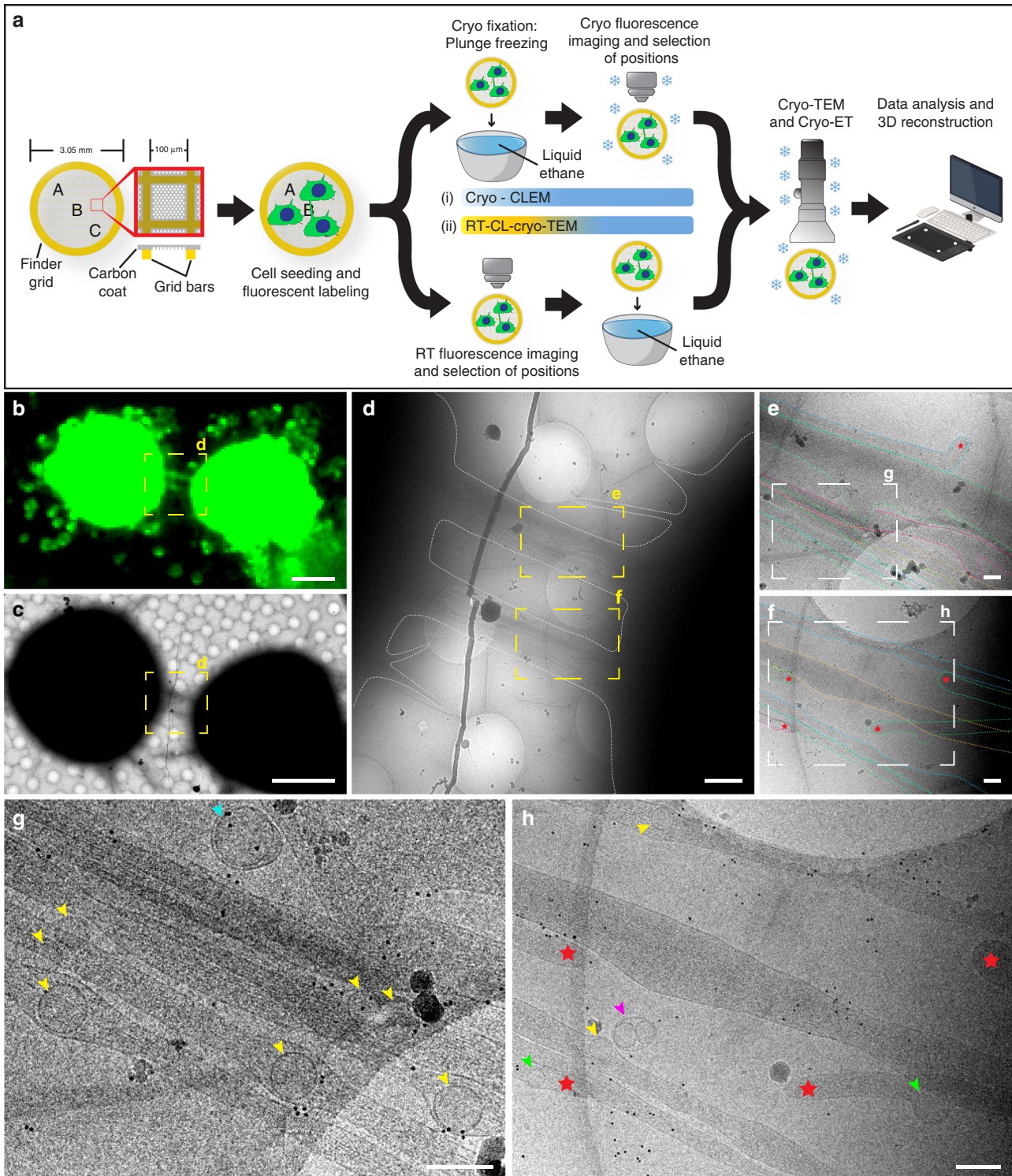

**Fig. 1** Correlated light and cryo-electron microscopy strategies reveal individual TNTs. **a** A schematic diagram of the experimental workflow and approaches used to observe TNT-connected CAD cells by cryo-TEM. **b** Cryo-FM image of two CAD cells connected by a TNT stained with WGA (green). **c** Cells in **b** observed in cryo-TEM at low magnification. Yellow dashed squares in **b**, **c** are shown at intermediate magnification in **d**. Yellow dashed rectangles in **d** are shown at high magnification in **e** and **f**, respectively. The plasma membrane of individual TNTs (iTNTs) were drawn with dotted colored lines in **e** and **f** to show membrane boundaries. Enlargements of white dashed rectangles in **e** and **f** are shown in **g** and **h**, respectively. **g**, **h** Various vesicular compartments were observed within and between iTNTs: yellow arrowheads show single vesicles; pink arrowheads show vesicles surrounded by an outer membrane; turquoise arrowheads show vesicles enclosed by a double membrane. Red stars in **e**, **f**, and **h** show tips of iTNTs extending/retracting from neighboring cells. Green arrows indicate vesicles inside extending/retracting iTNT tips observed in **h**. Scale bars: **b**, **c**, 10 μm; **d**, 1 μm; **e**–**h**, 200 nm

Overall, these data support that iTNTs correspond to the FM definition of TNTs. However, the number of TNTs imaged by cryo-CLEM was not sufficient for quantitative analysis, therefore we established a variant workflow by chemically fixing CAD cells and imaging them by FM at room temperature prior to rapid freezing and cryo-TEM (ii-CLEM) (Fig. 1a). Importantly, cryo- and chemically fixed samples (i.e., i-CLEM vs. ii-CLEM) (Supplementary Fig. 1m–1p)[29] structural details looked indistinguishable.

**N-Cadherin decorates threads holding iTNTs in a bundle.** Our previous data demonstrated that TNTs and filopodia in CAD cells are regulated by the same actin modifiers but in an opposite manner[19]. Thus, to further increase the number of TNTs on TEM grids, we used CK-666, an Arp 2/3 complex inhibitor previously shown to inhibit filopodia[30], which, based on our hypothesis, should increase TNTs. Indeed, our quantitation of adherent filopodia and TNT-connected cells[19,31] showed that CK-666 inhibits the formation of adherent filopodia (Supplementary Fig. 3a, b), and increased the number of cells connected via TNTs (Supplementary Fig. 3c, d). TNTs in CK-666-treated CAD cells showed no detectable structural differences in cryo-TEM compared to the untreated control (Supplementary Fig. 3g–j). Importantly, we also show that CK-666-induced TNTs are functional (i.e., able to support transfer) by using an assay which consists of monitoring the transfer of DiD-labeled vesicles between a donor-acceptor cell co-culture[19,23,27] (Supplementary Fig. 3e, f).

Therefore, we used these conditions and cryo-ET to elucidate (i) the spatial and structural arrangement of iTNTs with respect to each other, (ii) the actin organization inside tubes, and (iii) better characterize the vesicular compartments located inside iTNTs. TNTs were comprised of 2 to 11 iTNTs with a bundle diameter between 145 and 700 nm (305 nm on average). Each iTNT had an average diameter of 123 nm (SD = 66 nm). While the majority of iTNTs (~95%) had a diameter of less than 200 nm (Fig. 2a), thicker iTNTs (up to 550 nm) were occasionally observed as part of a bundle of two or more iTNTs. The spacing between individual tubes ranged between 8 and 90 nm for parallel iTNTs, while for less parallel iTNTs the spacing increased to 200 nm. Ninety percent of these distances in parallel iTNTs ranged between 8 and 60 nm (Fig. 2b).

This peculiar arrangement prompted us to ask the question of how individual iTNTs (Fig. 2c–f and Supplementary Movie 1) could be held together and remain bundled to form this unique TNT structure. Along the regions imaged by cryo-ET, we observed thin structures connecting the plasma membrane of two or more iTNTs (turquoise arrows, Fig. 2g, h, and Supplementary Movies 2 and 3). We speculate that N-Cadherin, a transmembrane adhesion protein previously observed inside the TNTs of other cell types[32], could be responsible for the formation of these linkers. To test this hypothesis, we first confirmed the presence of N-Cadherin in TNTs of CAD cells using FM (Supplementary Fig. 4a). Next, we employed cryo-ET on N-Cadherin immunogold-labeled samples (Supplementary Movie 4). Interestingly, we observed that the thin linkers connecting iTNTs were positively labeled for N-Cadherin (green arrowheads, Fig. 3a–c and Supplementary Fig. 4b, c).

Furthermore, tomograms slices shown in Fig. 2g, i–k also suggested that iTNTs were held together by long threads that coiled around them (green arrows, Fig. 2g, k, (right), and Supplementary Movie 5). These threads were also observed in native frozen conditions, excluding the possibility they were merely chemical fixation artifacts. Of interest, we found N-Cadherin gold labeling at the contact points between these threads and the iTNT surface (Fig. 3b). Taken together, our data

suggests that N-Cadherin holds iTNTs together and presumably provides stability for the overall bundle of iTNTs.

Another open question in the field is how cargo moves inside TNTs. Vesicles inside iTNTs were typically located between the plasma membrane and the actin bundle of parallel filaments, which filled the entire volume of the tube. Because the vesicle size often exceeded the diameter of the individual tube, they frequently bulged out from the plasma membrane, squeezing the actin filaments (Figs. 1g, h, 2g, h, and Supplementary Fig. 2a–f). By cryo-ET, we observed thin electron-dense structures that presumably connected intracellular vesicles to actin filaments (pink arrowheads, Fig. 2g), and short (~10 nm) spike-like structures that seemed to link vesicles to the plasma membrane on the opposite side of the actin bundle (orange arrows, Fig. 2g).

These observations are consistent with the hypothesis that vesicles move inside TNTs on actin filaments using myosin motors[1,27] and with the fact that different myosin motors have been found in TNTs by FM[1,5,33–35]. We decided to investigate the localization of one unconventional myosin motor, Myo10, as it was shown to increase TNT number and transfer of vesicles when overexpressed in CAD cells[27]. By applying our correlative approach (ii-CLEM) on GFP-Myo10-transfected cells, (Fig. 3d), we located Myo10 inside TNTs and associated to vesicular compartments. (Fig. 3e–h and Supplementary Movies 6 and 7). While the mechanism by which Myo10 increases TNT formation needs to be further investigated, the co-localization of Myo10 with vesicular structures inside iTNTs supports its role in motoring cargo on actin filaments inside iTNTs[27].

Another related question is whether transfer of vesicles is unidirectional or bidirectional. Within iTNTs, actin was organized in long bundles of filaments running parallel to each other (Figs. 2f–h, k, 3b, c, f, g, and Supplementary Movies 1–7). Due to beam damage, an inherent technical limitation of cryo-TEM, we were not able to image the actin bundle along the entire length of iTNTs. However, by imaging non-consecutive 1.2–1.5 μm-long regions of iTNTs, each iTNT contained an uninterrupted parallel actin bundle extending along the whole length of the area imaged. This strongly suggests that each iTNT contains a single continuous bundle of parallel actin filaments. Actin bundles filled the entire lumen of iTNTs with a diameter of <300 nm (Figs. 1g, h, 2f–h, k, and 4a) and although we could not address the polarity of the filaments we speculate that the actin bundle polarity remains the same within each iTNT but could feature opposite polarities in different, parallel, iTNTs (as shown in Fig. 1h). Tracing plot profiles and performing Fourier power spectra analysis on our cryo-CL-ET estimated the average distance between the center of adjacent actin filaments in bundles at 10 nm (SD = 0.8 nm) and 4.7 nm (SD = 1.1 nm) between their surfaces (Fig. 4a–f and Supplementary Movie 8).

We further examined cross-sections of iTNTs and found that actin filaments inside iTNTs were arranged in hexagonal arrays, with a 9.9 nm distance between the center of adjacent actin filaments (SD = 1.6 nm), and 5.1 nm (SD = 1.4) between their surfaces (Fig. 4g, h).

**Filopodia show different structural features compared to TNTs.** The structures we described above as TNTs appeared different compared to filopodia previously studied by cryo-ET[36,37] for two main reasons: (i) filopodia are usually isolated and do not run in inter-connected bundles; (ii) filopodia do not contain vesicular structures or organelles. However, filopodia were never studied in CAD cells at the ultrastructural level. Thus, we decided to use our cryo-EM approach to analyze the structure of filopodia in CAD cells to directly compare them with TNTs. To increase filopodia number on TEM grids, CAD cells were transfected with the vasodilator-stimulated phosphoprotein

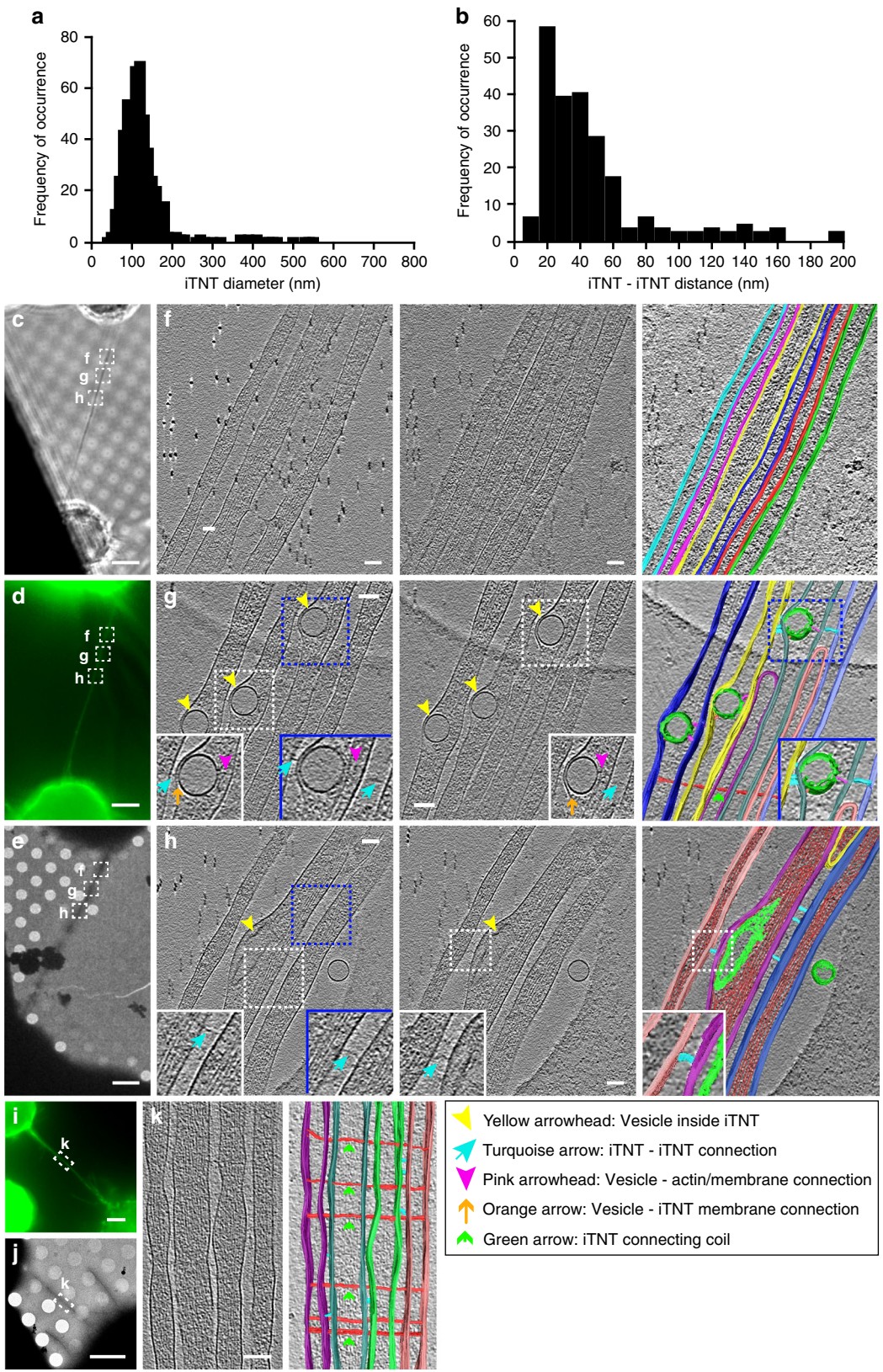

Yellow arrowhead: Vesicle inside iTNT
Turquoise arrow: iTNT - iTNT connection
Pink arrowhead: Vesicle - actin/membrane connection
Orange arrow: Vesicle - iTNT membrane connection
Green arrow: iTNT connecting coil

(VASP), a protein previously shown to be an effective inducer of filopodia in different cells[12,19,27]. Importantly, filopodia induced by VASP looked indistinguishable from those of untreated cells at the ultrastructural level (Fig. 5a–i).

As previously reported for other cell types[36,37] we did not observe vesicles or other organelles inside the filopodia of CAD cells. Furthermore, unlike TNTs, filopodia in CAD cells were never in bundles. Instead, they were single isolated protrusions,

**Fig. 2** Imaging iTNTs in 3D using correlated light and cryo-electron tomography. **a** iTNT diameter distribution in nm. **b** Distribution of the distance between parallel and non-parallel iTNTs in nm. CAD cells connected by a TNT stained with WGA (green) imaged by phase contrast (**c**), epifluorescence (**d**), and low-magnification TEM (**e**). Dashed squares over the TNT in **c–e**, denoted **f–h**, correspond to non-consecutive high-magnification cryo-ET slices (**f–h**, left, middle, and Supplementary Movies 1–3). **f–h** (right) Rendering of tomograms (Supplementary Movies 1–3, respectively). Turquoise arrows in **g** and **h**, left, middle show filaments connecting iTNTs. Vesicles within iTNTs (yellow arrowheads, **g** and **h**, left, middle) use thin filaments to connect to the plasma membrane on one side (orange arrows, **g** left, middle) and actin on the other (pink arrowheads, **g** left, middle). Additional example of cells connected by a TNT is shown in **i** (epifluorescence) and **j** (low-magnification TEM). Region denoted by a white dashed rectangle in **i–j** is shown at high-magnification cryo-TEM in **k** left. **k** right, rendering of **k**, left (Supplementary Movie 5). Thin filaments connect the bundle of iTNTs by coiling around them (green arrows, **g** and **k** (rendering)). Source data for **a**, **b** are provided as a Source Data file. Scale bars: **c–e**, **i–j**, 5 μm; **f–h**, **k**, 100 nm

(straight, bent, or twisted around other single filopodia (Fig. 5b, c, e, f, h, i, k–m and Supplementary Movie 9–12)) with an average diameter of 174.9 nm, ranging between 118.3 and 285.5 nm.

As previously described for other neuronal cell lines[37], actin filaments were mostly organized in tight parallel bundles that extended into the tip of the filopodium (Fig. 5b, c, e, f, h, i, k–o, and Supplementary Movies 9–12). Although the average diameter of filopodia is within the range of the average diameter of iTNTs (i.e., 174 vs. 123 nm, respectively), differently from TNTs, filopodia's actin filaments did not run uninterrupted along the whole length of the area imaged (1.2–1.5 μm). Instead, they were organized in bundles comprised of shorter filaments with a length varying between 300 and 1100 nm, with only 15% having a length longer than 1 μm (Fig. 5k–m). Alternatively, filopodia also displayed short parallel actin bundles (Fig. 5b, k–l, and Supplementary Movies 9 and 12) that intermingled with short-branched filaments (red arrowheads, Fig. 5b, h, and Supplementary Movies 9 and 11). This latter arrangement was never observed in TNTs, but was similar to what was previously shown for filopodial conformations in *Dyctostelium discoideum* amoebae[36].

In agreement with results reported by Aramaki et al.[37], in a different neuronal cell line, the average distance between the centers of adjacent actin filaments in bundles was 10.4 nm (SD = 0.4 nm) and 4.7 nm (SD = 0.7 nm) between their surfaces (Fig. 5o–r). This is similar to the estimate obtained for actin bundles within iTNTs (Fig. 4c–e), suggesting that although the overall actin configuration is different in iTNTs, the distance between parallel actin filaments in both structures is maintained. By observing cross-sections of filopodia, the actin bundle diameter ranged between 80 and 170 nm and the number of filaments per bundle ranged between 38 and 150. Moreover, actin filaments in filopodia were bundled by cross-linkers arranged in hexagonal arrays comprised of 12–30 filaments, with a ~4.7 nm distance between their surfaces (Fig. 5 k–s). A similar actin-bundling hexagonal pattern was previously shown to be mediated by the actin-binding protein fascin[37].

**TNTs in SH-SY5Y cells are made of functional iTNTs**. While the data presented above unequivocally demonstrates that TNTs are specialized structures in CAD cells, one of the issues in the field of TNTs is the heterogeneity of TNT-like structures observed in different cell types. Due to the diffraction limit of resolution in FM, it is not clear whether the heterogeneity in TNTs reflects a real ultrastructural difference, or whether different cellular features are being reported and are all named TNTs, although possibly referring to different structures. To confirm that the structural characteristics of TNTs described above are not cell-type specific, a neuronal cell model of human origin, SH-SY5Y cells, previously shown to connect via TNTs using FM, was investigated next[4,24,25] (Supplementary Fig. 5a). To investigate the nature of TNTs connecting SH-SY5Y cells we first analyzed them by SEM. Similar to CAD cells, thicker TNTs appeared to endure harsh classic EM sample preparation steps (red

arrowhead, Supplementary Fig. 5b), while thinner structures would often break (yellow arrowheads, Supplementary Fig. 5c, d, e). Compared to CAD cells, however, more thin structures connecting SH-SY5Y cells survived classical embedding conditions (blue arrowheads, Supplementary Fig. 5e). By employing CLEM approaches on this cell model, TNTs connecting SH-SY5Y cells were also comprised of bundles of two or more iTNTs (Fig. 6a–e) while single thick connections were occasionally observed (Supplementary Fig. 5f–i). iTNTs connecting SH-SY5Y cells had an average diameter of 120.7 nm (SD = 71.4 nm) (Fig. 6f) and contained vesicular compartments (yellow arrowheads, Figs. 6d–e), with an average diameter of 104 nm (SD = 57.8 nm) (Fig. 6g). Importantly, control experiments demonstrated that iTNTs connecting SH-SY5Y cells fixed by rapid freezing were virtually identical to those chemically fixed, as in CAD cells (Supplementary Fig. 5j–m).

In order to demonstrate that the structures identified by cryo-CLEM in SH-SY5Y cells were functional, we examined whether they also transferred cargos. In addition to vesicles, TNTs are able to transfer mitochondria between connected cells[4,38–40]. Compared to DiD-labeled vesicles, mitochondria are particularly bright when fluorescently labeled and, therefore, well suited to be imaged by live microscopy[41,42]. When cells labeled with WGA (green) and MitoTracker (red) were recorded by live cell imaging, mitoTracker-positive puncta moved inside a TNT in a unidirectional fashion at an average velocity of 0.05 μm/s (SD = 0.03 μm/s) (Fig. 6h and Supplementary Movie 13). As previously reported for PC12 cells[41], mitochondria did not move through the TNT at a uniform speed and seemed to accelerate in segments where the TNT appeared to be completely straight. As exemplified in Supplementary Movie 14, labeled mitochondria traveled all along an iTNT and accumulated in the cytoplasm of the neighboring cell (blue arrowhead, Supplementary Fig. 5n and Supplementary Movie 14).

The presence of mitochondria inside iTNTs was confirmed at the ultrastructural level by ii-CLEM and cryo-ET. By cryo-ET, the mitotracker labeled structures seen by FM (Fig. 6i–k) were mitochondria based on their size, shape, and presence of mitochondrial cristae (Figs. 6m–o and Supplementary Movie 15). Intriguingly, mitochondria were only observed in one of the iTNTs. As with vesicles, mitochondria also created a bulge in the iTNT containing it (Fig. 6l, n, o, and Supplementary Fig. 5o–s). Altogether, this data demonstrates that the structure of functional TNTs (i.e., allowing transfer of cargoes inside their lumen) is similar between CAD and SH-SY5Y cells, two different neuronal cell models from different origins and species.

**Open-ended-TNTs connect the cytoplasm of cells**. Given that vesicles and larger organelles such as lysosomes and mitochondria can transfer via TNTs[4,5,7], it is conceivable that TNTs are required to be open at the cell contact site. This notion, however, remains controversial, as it has not been sufficiently supported by ultrastructural data[1,11,15]. Cryo-ET was not suitable to address

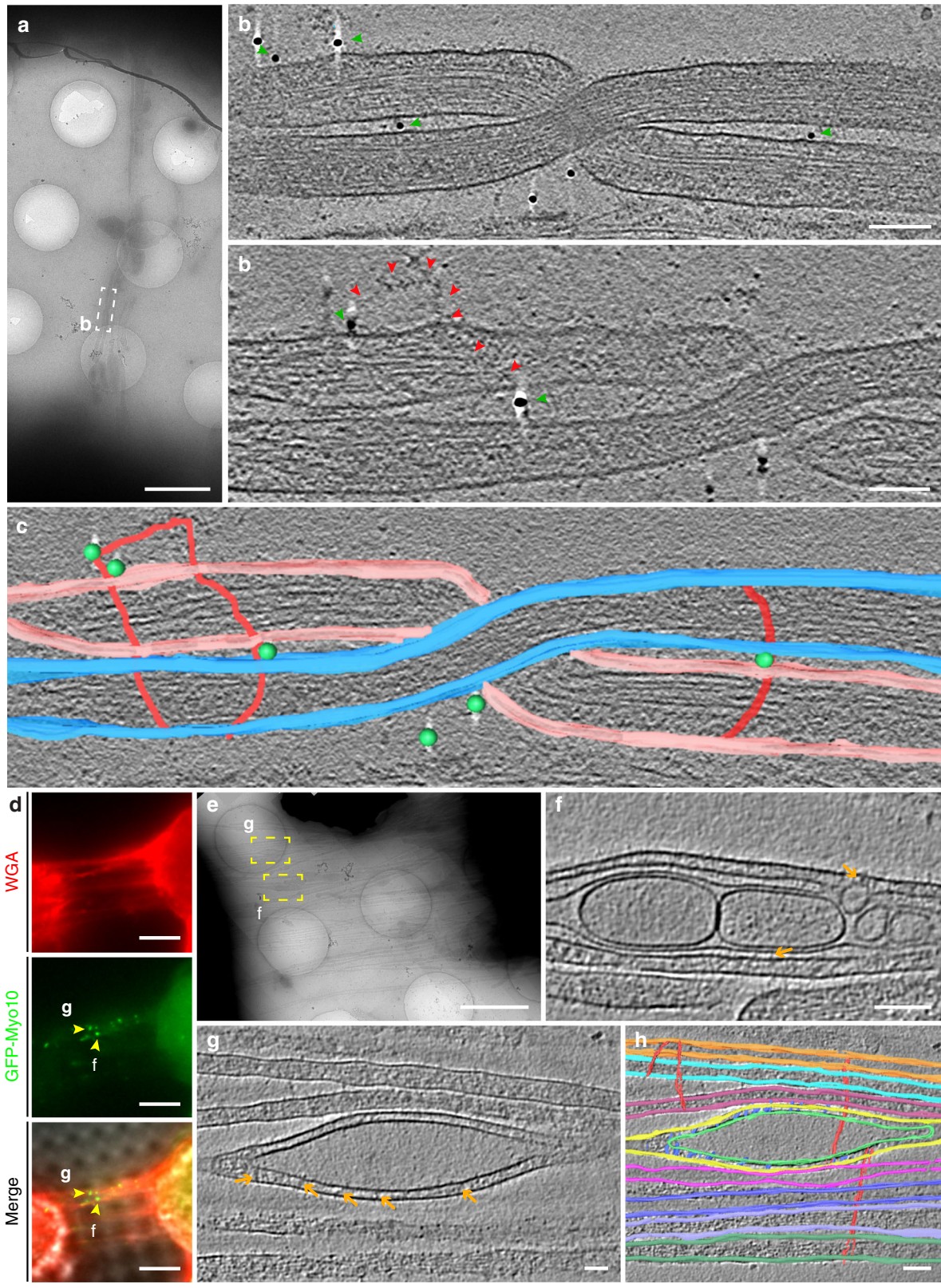

this question as the connecting regions between TNTs and cell bodies were too thick (>500 nm in thickness) to be investigated by TEM (Supplementary Fig. 6a–c). To generate 3D EM images that enable the analysis of TNT-to-cell contact sites we therefore employed correlative focus ion beam SEM (FIB-SEM) tomography on both CAD and SH-SY5Y cells.

TNTs connecting CAD cells were first identified by FM (Fig. 7a) and subsequently imaged by FIB-SEM. Under the imaging conditions used, the resolution of our tomograms was sufficient to discern plasma membrane boundaries of cells and TNTs. Manually segmented tomogram renderings (right panel, Fig. 7a) and volume renderings created with the Amira

**Fig. 3** Ultrastructural analysis of N-cadherin and Myo10 in iTNTs. **a** Low-magnification electron micrograph displaying N-Cadherin immunogold-labeled CAD cells connected by iTNTs. (**b**-top) White dashed rectangle in **a** correspond to two high-magnification non-consecutive cryo-tomogram slices (25 nm in thickness) shown in **b**-top and **b**-bottom (Supplementary Movie 4). Green arrowheads in **b** and spheres in **c** indicate 10 nm gold particles attached to a polyclonal secondary antibody that binds to an N-Cadherin primary antibody. Red arrows indicate a long thin thread filament surrounding iTNTs anchored to the membrane of iTNTs by N-Cadherin molecules (green arrows). **c** Segmentation rendering of the tomogram described in **b**; N-Cadherin (green beads), thin thread surrounding iTNTs (green arrowheads). **d** Epifluorescence micrographs of CAD cells overexpressing GFP-Myo10 connected by TNTs stained with WGA (red). Yellow arrowheads indicate GFP-Myo10 signal in TNTs. **e** Low-magnification electron micrograph corresponding to TNTs shown in **d**. **f**, **g** High-magnification cryo-tomography slices corresponding to the yellow dashed squares in **e**. Yellow arrowheads in **d** mark GFP-Myo10 vesicles (Supplementary Movies 6 and 7). Orange arrows indicate vesicle-iTNT connections. **h** Rendering of tomogram shown in **g** (Supplementary Movie 7). Scale bars: **a**, **e**, 2 μm; **b**, **f**, 100 nm; **d**, 5 μm; **g**, **h**, 50 nm

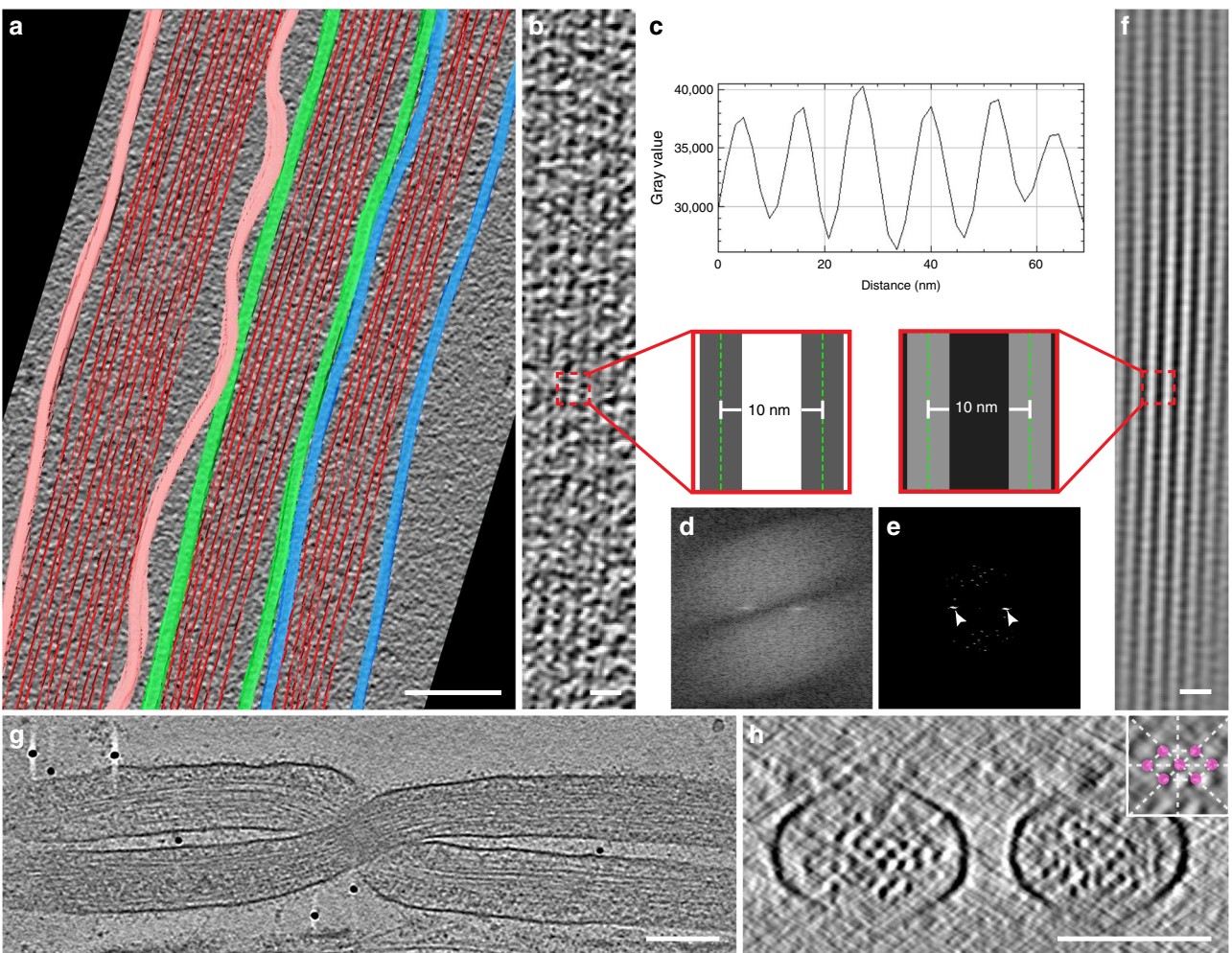

**Fig. 4** Cryo-electron tomography reveal F-actin organization in iTNTs. **a** Segmentation rendering obtained from a tomogram showing iTNTs (Supplementary Movie 8). The average size and spacing between filaments of an iTNT shown in **b** were measured by computing the plot profile of actin bundles residing within. **c**–**e** Filament-to-filament distance measured by extracting peaks in the frequency domain using the fast Fourier transform (white arrowheads, **e**) and by computing plot profiles (**c**). **f** Model of the actin filaments displaying a parallel bundle in iTNTs obtained by retransforming peaks in the frequency domain shown in **e**. Insets in **b** and **f** show distances between the average measurements in nm obtained by the analysis shown in **c** and **d**. **g** High-magnification cryo-tomogram slice of two iTNTs (Fig. 3b and Supplementary Movie 4). **h** Cross-section of the cryo-electron tomogram in **g** displaying the actin arrangement within two iTNTs. Scale bars: **a**, 100 nm; **b**, **f**, 20 nm; **g**, **h**, 100 nm

software (Fig. 7b), helped with the visualization of contact sites (Fig. 7c) in 3D (Supplementary Movie 16) showing fusion of TNTs with the membrane of the cell bodies of both connected cells (orange and pink stars, Fig. 7b, c). Three-dimensional tomogram volumes revealed a TNT comprised of three iTNTs with an average diameter of 108 nm, creating a bundle of 388 nm in diameter.

FM-identified TNTs connecting SH-SY5Y cells (Fig. 7d) imaged by FIB-SEM showed similar features as those shown for CAD cells. Three-dimensional segmentation renderings (Fig. 7d), volume renderings (Fig. 7e), and careful observation of contact sites (orange and pink stars, Fig. 7f), revealed an open-ended TNT comprised of two iTNTs with an average diameter of 235 nm (Supplementary Movie 17).

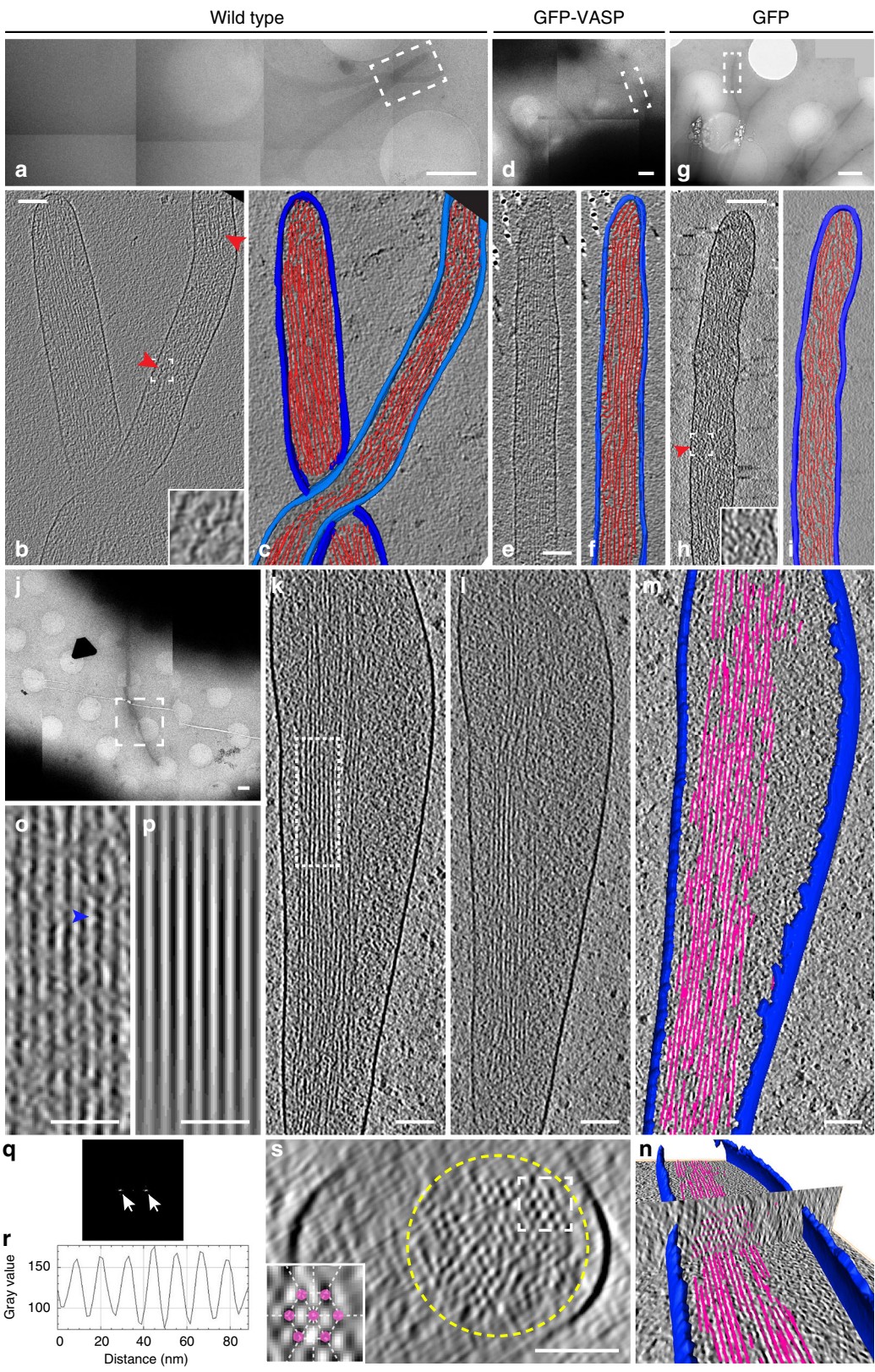

In a separate example, we observed a pair of TNTs of 220 nm in diameter each connecting two SH-SY5Y cells by FM (Supplementary Fig. 7a). One of these tubes was inserted inside the apposing cell and appeared to be closed at its tip (red arrowhead, Supplementary Fig. 7a, b, Supplementary Movie 18).

This invaginating configuration (3D volume reconstruction shown in Supplementary Fig. 7c), could presumably be the result of a pre-TNT fusion event. Alternatively, this observation could indicate that TNTs can either be open- or close-ended at contact sites.

**Fig. 5** Cryo-electron tomography reveal actin ultrastructure in filopodia. Representative electron micrographs of filopodial protrusions of untreated (**a–c**, **j–m**), GFP-VASP (**d–f**), and GFP (control) (**g–i**) transfected CAD cells imaged by cryo-TEM at low magnification (**a, d, g, j**). **b, e, h** Slices from volumes rendered in **c, f,** and **i** correspond to white dashed rectangles shown in **a, d,** and **g**, respectively. Red arrowheads in **b** and **h** indicate branched actin configuration. **k–l** slices corresponding to white dashed square in **j. m–n** Three-dimensional automated rendering of **k–l** and cross-section of the filopodium (**n**). Segmentation of the plasma membrane (blue) and actin (pink). **o** High-magnification view of filaments corresponding to the white dashed rectangle in **k**, where filaments display a wavy periodic pattern (blue arrowhead, **o**). By computing its plot profile (**r**), the average filament size and spacing between filaments was measured. **p** Model of the parallel filament bundle obtained from the peak extraction in the fast Fourier transform (**q**). **s** Cross-section through the cryo-electron tomogram of the white dashed rectangle in **j**. Scale bars: **a, d, g, j,** 1 μm; **b, c, e, f, h, i, k, l, m, n,** 100 nm, **o, p,** 50 nm

It should be noted that TNTs of both CAD and SH-SY5Y cells in our FIB-SEM volumes did not display the straight and smooth morphology typically observed by FM and cryo-EM. We believe tubes can deform during sample preparation or milling. Nonetheless, this demonstrates that TNTs observed by FM can be open-ended at TNT-to-cell contact sites and thus directly link the cytoplasm of two connected cells.

## Discussion

TNTs have been shown to play a vital role in the intercellular spreading of various cargos, which include deleterious materials such as amyloidogenic proteins and have therefore been implicated in many different physiological and pathological processes[2–5,22].

While the morphological features of TNTs have been extensively described in various cell types using fluorescence microscopy, to date, no reliable EM method has been applied to TNTs that would allow their observation at the nanoscale resolution without jeopardizing their fragile membranous and cytoskeletal composition. Indeed TNTs have only been studied at the ultrastructural level by conventional SEM and TEM methods in a handful of publications[1,15–18], and no study has attempted to assess the structure of TNTs in correlative mode.

Here, we set-up a workflow for correlative light- and cryo-ET microscopy that, to the best of our knowledge, is the only approach to identify and characterize the underlying structures behind TNTs observed by FM. By using this approach, we were able to observe TNTs connecting two different neuronal cell models, CAD and SH-SY5Y cells, at the nanometer resolution.

Our results show that most TNTs, which appear as single connections by fluorescence microscopy, are in fact made up of several individual tunneling nanotubes (iTNTs). iTNTs often run together in a parallel fashion but occasionally braid over each other. We speculate that this conformation might greatly increase their stability and elasticity, allowing them to withstand movements between connected cells. In addition, our tomograms suggest the existence of thin linkers that connect adjacent iTNTs (Figs. 2, 3, and 8). We identified N-Cadherin, a transmembrane cell adhesion molecule, as a component of these connections. We speculate that these connections could serve as guidance for growing iTNTs to run on top or beside of one another and therefore be of importance during TNT formation (see below). These contacts could also increase the mechanical stability of the overall bundle by holding iTNTs together. Furthermore, tomogram reconstructions suggested that long threads that coiled around iTNTs held the bundle together. Interestingly, N-Cadherin was also found at the base of these threads, connecting them to the membrane of iTNTs. While pointing to an important role in TNT formation in neuronal cells, our N-Cadherin data paves the way for additional studies required to understand its specific role, and to uncover the nature and assembly of these filaments.

Each iTNT contains one actin bundle organized in a highly ordered fashion and likely extend all along the tube. In most cases, the actin bundle filled the entire lumen within iTNTs. This suggests the same actin polarity in a single tube. We also observed tips of iTNTs likely in the process of extending to- or retracting from either of the two connected cells (Figs. 1 and 6). We hypothesize that this event could represent a 'snapshot' of the mechanism by which TNTs are formed and maintained, a highly dynamic process that involves multiple iTNTs extending and retracting between opposite cells. We also speculate that iTNTs extending from opposing cells would have opposite polarities, thus allowing unidirectional transfer in opposite directions. In support of this hypothesis we could observe vesicles inside distinct iTNTs (within the TNT bundle) that appear to originate from opposite sides (Figs. 1h and 8). This could explain bidirectional vesicular transfer previously observed inside what appeared as a single TNT by FM[40]. Further studies combining live microscopy with cryo-TEM will be required to directly confirm these hypotheses. Alternatively, one could imagine that each iTNT could contain actin filaments with opposite polarity, or both actin and microtubules. While our tomograms are suggestive of a single actin bundle per iTNT, the presence of tubulin has been observed (by FM) in thicker connections, mainly in cells of the immune system, but not inside TNTs connecting neuronal cells[1,43,44]. We speculate that TNTs containing tubulin might be different from the ones we observe here in neuronal cells. Furthermore, there is no evidence that TNT-like structures in immune cells are open-ended[15], therefore it will be interesting to inquire their ultrastructure by correlative EM.

One distinguishing feature of TNTs is that they allow cargo transfer. We detected vesicular compartments of different shapes and sizes inside the single iTNTs (Supplementary Fig. 2), reinforcing the wealth of literature demonstrating the intercellular transfer of vesicles and organelles via TNTs[2,4,5]. In addition, we observed extracellular vesicles surrounded by a double membrane, which could have derived from either budding or fission from an iTNT. Exosome markers have been found in TNTs. They have also been proposed to travel inside TNTs and shown to stimulate TNT formation[45,46]. We found examples of multivesicular body-like structures inside iTNTs and external vesicles bearing a single-membrane, which could be exosomes. However, future studies will be necessary to investigate the nature of these structures.

Nonetheless, by correlative cryo-TEM we clearly identified mitochondria inside iTNTs connecting SH-SY5Y cells (Fig. 6). Altogether with live microscopy data showing mitochondria traveling through TNTs, these findings indicate that mitochondria can be transported within iTNTs by a microtubule-independent mechanism (Fig. 6n, o). Interestingly, while microtubules and their motors have been established as important factors for mitochondrial transport[47], emerging evidence indicates that mitochondria interact with the actin cytoskeleton in many cell types[48–50]. Although we could track single mitochondria and measure their velocity by live imaging, the motors involved in their transport through iTNTs, remain unidentified.

On the other hand, by overexpressing Myo10, previously shown to increase TNT formation in CAD cells[27], we were able to

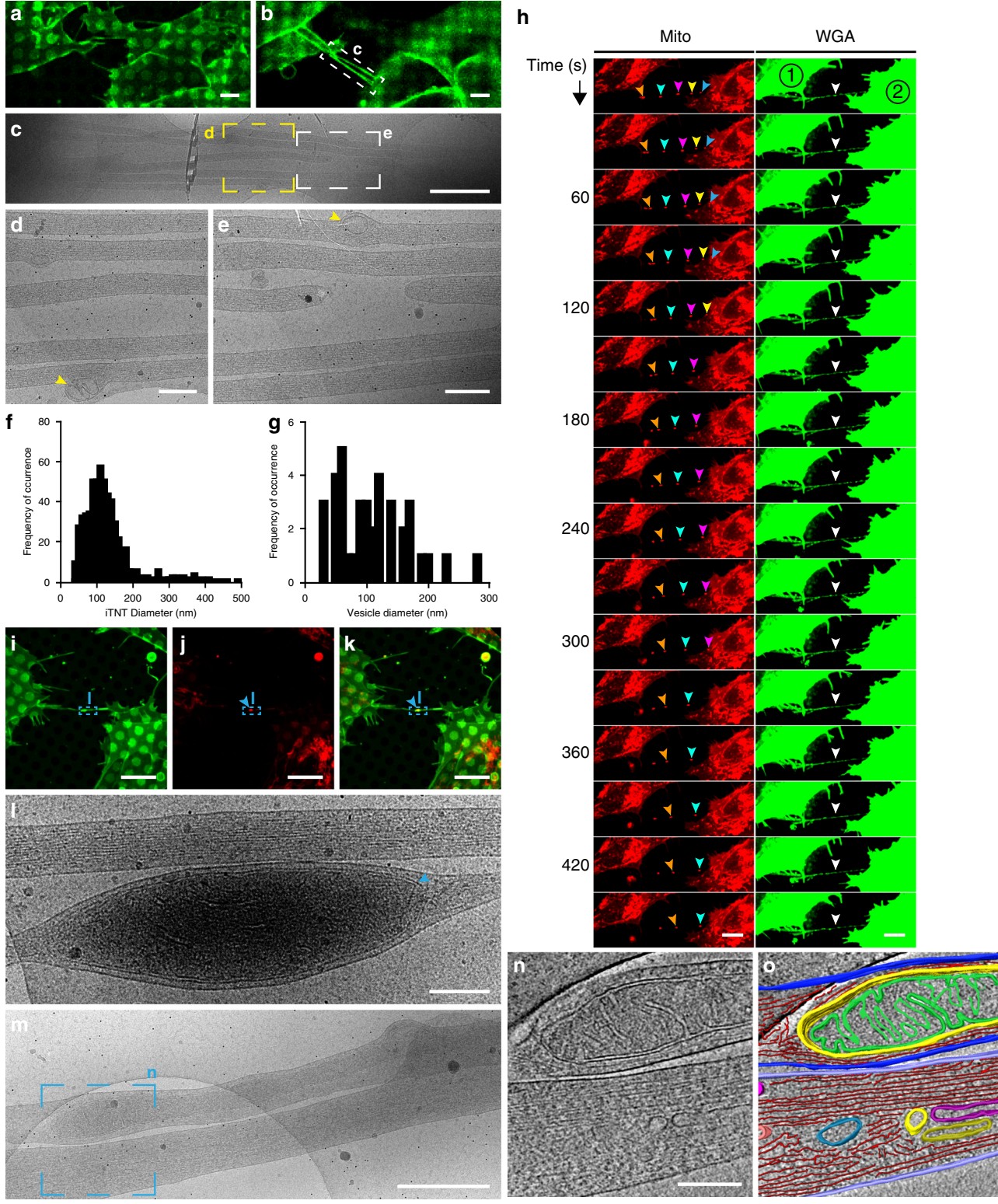

localize this actin driven motor with vesicular structures inside the lumen of iTNTs. While the mechanism of cargo movements on actin bundles needs to be further enquired, and could involve other motors previously observed in TNTs in FM[1,27], these data support a role for Myo10 in this process.

Overall, our data indicate that TNTs are largely different from filopodia. Tomograms revealed that although the average diameter of filopodia is within the range of the average diameter of iTNTs (174 vs. 123 nm), similar to other cells, filopodia in CAD

cells are isolated protrusions, not found in bundles. Importantly, as previously shown in other cell types[36,37], filopodia in CAD cells do not appear to contain vesicles. Finally, filopodia displayed various actin arrangements compared to TNTs. While the actin in TNTs was always found in parallel bundles that ran along the entire length of the area imaged (up to 1.5 µm) and continued seemingly uninterrupted in the next (non-consecutive) area imaged; in filopodia, tight parallel bundles were of much shorter length (with only 15% having a length longer than 1 µm),

**Fig. 6** Structural analysis and live imaging reveal mitochondrial transport via iTNTs. **a** Confocal micrograph of SH-SY5Y cells stained with WGA (green). **a** Lower and **b** upper optical slices show TNTs hovering over the substrate. **c** Low-magnification electron micrograph corresponding to white dashed rectangle in **b**. **d**–**e** High-magnification micrographs corresponding to yellow and white dashed squares in **c**. Yellow arrowheads indicate membranous compartments inside iTNTs. **f**–**g** Diameter distribution of iTNTs (**f**) and vesicles found within them (**g**) in nm. **h** Time-lapse sequence of two SH-SY5Y cells stained with WGA (green) and MitoTracker (red) reveal mitochondrial puncta (orange, cyan, pink, yellow, and blue arrowheads) traveling across a TNT (white arrowhead) entering the apposing cell, (cell #2). **i**–**k** SH-SY5Y cells stained with WGA (green) and mitotracker (red), (blue arrowheads indicate mitochondria inside the TNT). **l** Micrograph corresponding to blue dashed rectangles in **i**–**k** reveal mitochondria (blue arrowhead) observed by fluorescence (**i**–**k**). **m** Micrograph displays two iTNTs, one containing mitochondria. **n** Tomogram slice corresponding to **m** (blue dashed square) reveal mitochondrial cristae, actin, and membranous compartments found inside iTNTs. **o** Rendering of **n**. Source data for **f**–**g** are provided as a Source Data file. Scale bars: **a**–**b**, 5 μm; **c**, 1 μm; **d**, **e**, **l**, **n**, 200 nm, **h**–**i**, 10 μm; **m**, 500 nm

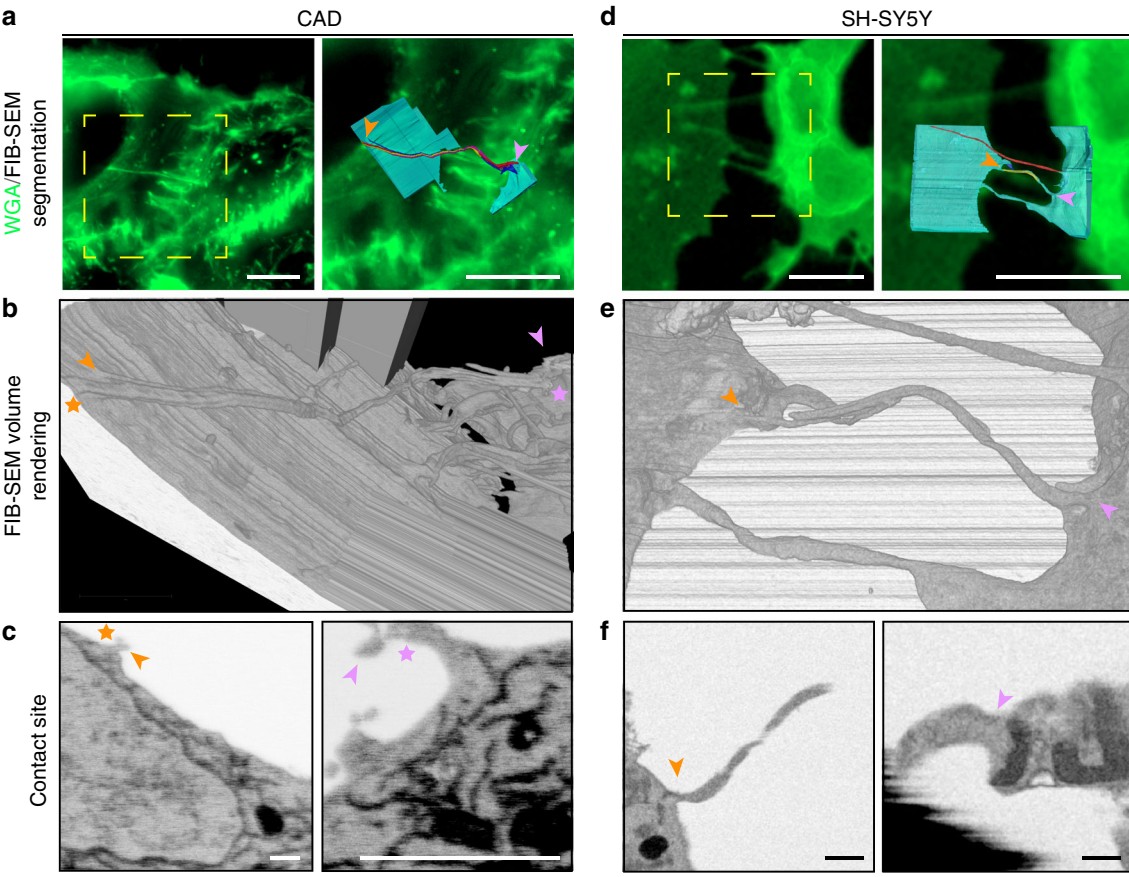

**Fig. 7** FIB-SEM of TNTs in CAD and SH-SY5Y cells open-open contact sites. **a**, **d** Confocal micrographs of TNT-connected CAD (**a**) and SH-SY5Y (**d**) cells stained with WGA (green). Yellow dashed boxes in **a** and **d** (left) show the regions of interest chosen for FIB-SEM volume acquisition. A magnified view of these regions is shown in **a** and **d** (right) overlaid with their FIB-SEM segmented rendering counterpart. Segmented cell bodies and single open-ended connections are shown in cyan. Open-ended iTNTs are shown in blue, yellow, and magenta. Single close-ended protrusions are shown in red. Orange and pink arrowheads indicate contact sites with cells. **b**, **e** Renderings of FIB-SEM volumes produced by the Amira software are shown for CAD (**b**) and SH-SY5Y (**e**) cells. **c**, **f** Single frames obtained from the raw FIB-SEM data of CAD cells (**c**), obtained at a pixel size of 2.5 nm and displayed in an x-y orientation, and SH-SY5Y cells (**f**) obtained at a pixel size of 10 nm and displayed in an *x-z* orientation. Panels show TNT-to-cell contact sites at both ends of the connection: orange arrowhead (left); pink arrowhead (right). Scale bars: **a**, **d**, 10 μm; **c**, **f**, 500 nm

arranged in parallel bundles intermingled with short-branched filaments. Interestingly, when we looked at the arrangement of parallel bundles in TNT and filopodia by cross-section analysis, we revealed similar arrangements, (i.e., similar distance between the center and surface of filaments). In filopodia we also showed a typical actin-bundling hexagonal pattern possibly due to the actin-binding protein Fascin[37]. A similar hexagonal pattern was observed in TNTs but under the imaging conditions used, the number of filaments per bundle could not be clearly detected for TNTs. This is possibly due to the lack of resolution achievable by our cryo-TEM on thick samples and the considerable technical challenges we faced when imaging iTNT bundles under cryogenic

conditions compared to isolated filopodia. Thus, to further analyze the structural differences and similarities in the actin arrangements within iTNTs, the use of TEMs with improved optics will be required.

To specifically address the controversial question in the TNT field whether TNTs are connecting the cytosol of two cells, we performed FIB-SEM tomography on CAD and SH-SY5Y cells, which allowed us to image the ends (or contact sites) of TNTs. Interestingly, FIB-SEM enabled us to identify TNTs open at both ends, in both cell lines. TNTs appeared to fuse with the membrane of cell bodies at separate locations, while in other cases they seemed to merge into one stem that fused with the cell. This

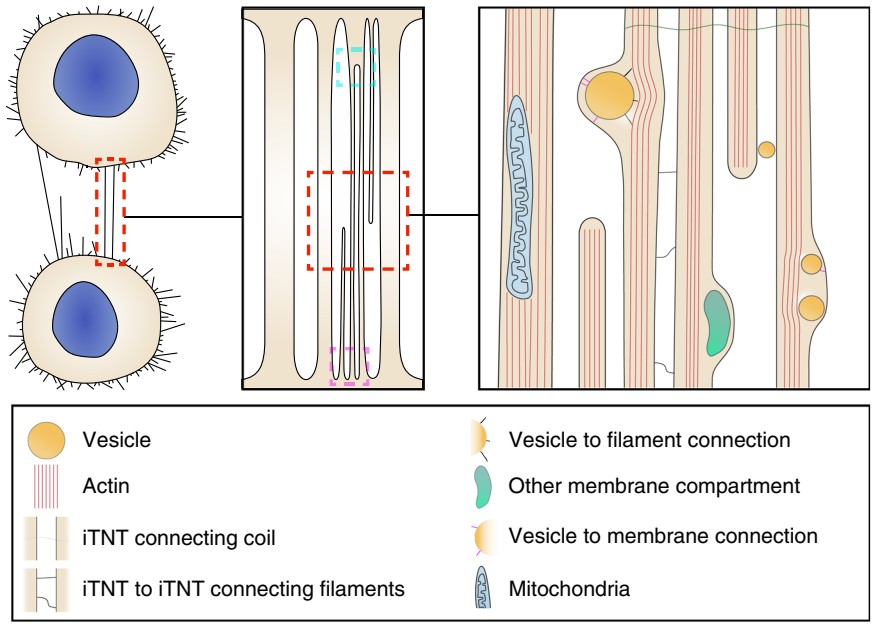

**Fig. 8** Schematic diagram depicting how cells connect via TNTs. TNT-connecting cells can either be single thick connections or a bundle of thin individual TNTs (iTNTs). iTNTs contain vesicles and mitochondria. Membranous compartments within iTNTs appear to be connected by thin filaments to actin on one side and to the inner side of the plasma membrane on the other. Thin membrane threads coil between and around several iTNTs. Cyan and magenta dashed squares show two types of contact sites: merging of iTNTs prior to fusion and fusion of iTNTs at separate locations, respectively

observation is particularly interesting as it suggests that iTNTs can also merge. Whether this occurs only at the ends or along their extension, or whether N-Cadherin is involved in this process, is an interesting question that needs to be further addressed.

We also found some close-ended connections and invaginations at contact sites. At this point, we cannot discriminate whether the differences observed are the result of the existence of distinct TNTs or whether it is due to temporal pre or post fusion events. Nonetheless, these data demonstrates that open-ended TNTs exist and could correspond to the functional TNTs structures observed by FM.

Overall, the data presented here provides a structural characterization of functional TNTs previously observed by light microscopy in two different neuronal cell models. They demonstrate their specific identity, highlight the morphological differences with filopodia, show the actin architecture at high resolution and the presence of vesicular structures and mitochondria inside them.

While many more questions remain open these data also guide the establishment of a TNT inducing and imaging platform for investigating the structural mechanism underlying TNTs, as well as for revealing the specific role that membrane and cytoskeleton-associated proteins may play in the formation and function of this important biological process.

## Methods

**Cell preparation and transfections.** CAD (mouse catecholaminergic neuronal cell line, Cath.a-differentiated) cells were kindly given by Hubert Laude (Institut National de la Recherche Agronomique, Jouy-en-Josas, France) and cultured at 37 °C in Gibco Opti-MEM (Invitrogen), plus 10% fetal bovine serum and 1% penicillin/streptomycin[23]. SH-SY5Y cells (neuroblasts from human neural tissue) were cultured at 37 °C in RPMI-1640 (Euroclone), plus 10% fetal bovine serum and 1% penicillin/streptomycin. Transient transfections were performed with Lipofectamine 2000 (Invitrogen) in accordance with the manufacturer's instructions. CAD cells were transfected in T25 flasks with the appropriate constructs: GFP-tagged constructs, GFP-VASP, GFP-Myo10, or H2B-mCherry for 3 h in serum-free medium and incubated O/N with complete medium. The following day, transfected cells were plated on grids[19].

**Pharmacological treatments.** For pharmacological assays, confluent CAD cells were mechanically detached and counted; 100,000 cells were plated for 24 h on Ibidi μ-dishes (Biovalley, France). Cells were then treated with CK-666 (SML0006, Sigma), an Arp 2/3 complex inhibitor, which was made up as 75 mM stock solution in dimethyl sulfoxide (DMSO) and diluted in complete medium to a final concentration of 25 μM or 50 μM. DMSO-alone control treatments were performed as part of every experiment involving drug treatment. Cells were treated for 15 min at 37 °C before fixation and fluorescent labeling. Acceptor cells were plated in T25 flasks (Corning, USA) and transiently transfected with an H2B-mcherry vector after 24 h. The following day, donor cells were detached and resuspended in complete medium with 1 μM of DiD-labeled vesicles and incubated for 30 min at 37 °C. Donor cells were then washed with complete culture medium, mixed with acceptor cells (1:1), and plated on 35 mm Ibidi μ-Dish (Biovalley, France). After an incubation of 1 h at 37 °C, cells were treated with CK-666 (25 μM and 50 μM) and DMSO (control) for 6 h. After washing cells with fresh, complete medium they were incubated at 37 °C O/N. The next day, cells were washed with phosphate-buffered saline (PBS), fixed in 2% PFA + 0.05% GA in 0.2 M Hepes for 15 min followed by fixation in 4% PFA in 0.2 M Hepes for 15 min, stained with DAPI (1:1000), and mounted with Aqua-Poly/Mount (Polysciences, Inc.). Images were acquired on an LSM 700 confocal microscope (Zeiss) with a 40X objective. Semi-automated detection and quantification of the number of DiD-labeled puncta was assessed with the open source software, ICY[7].

**Immunofluorescence labeling.** CAD cells plated overnight were washed carefully, fixed with 4% paraformaldehyde (PFA) for 15 min at 37 °C, quenched with 50 mM NH4Cl for 15 min, permeabilized and blocked with 0.075% Saponin in PBS containing 2% BSA (w/v) for 1 h at 37 °C. Cells were then incubated with a mouse anti-vinculin primary antibody (V9264, Sigma) (1:500) in 0.01% saponin and 2% BSA (w/v) in PBS. Cells were thoroughly washed and incubated for 40 min with a goat anti-mouse AlexaFluor 488-conjugated secondary antibody (Invitrogen) at 1:500 in 0.01% saponin and 2% BSA (w/v) in PBS. Cells were then carefully washed in PBS and labeled with WGA-647 (1:300 in PBS) for 5 min at room temperature (RT), and 5 min with Phalloidin-Rhodamine (1:300 in PBS); nuclei were labeled for 5 min with DAPI (1:1000). Cells were washed and mounted with Aqua-Poly mount (Polysciences, Inc.).

**Cell preparation for cryo-EM.** Carbon-coated gold TEM grids (Quantifoil NH2A R2/2) were glow-discharged at 2 mA and 1.5–1.8 × 10⁻¹ m bar for 1 min in an ELMO (Cordouan) glow discharge system. Grids were sterilized under UV three times for 15 min at RT and incubated at 37 °C in complete culture medium. Grids were carefully washed twice in PBS and coated with 40 μg/mL of fibronectin for 20 min at 37 °C. After three PBS washes, 250,000 CAD or 700,000 SH-SY5Y cells were seeded on grids and incubated O/N at 37 °C, resulting in 3 to 4 cells per grid square. Prior to chemical or cryo-fixation, cells were labeled with WGA-Alexa-488 (1:300 in PBS) for 5 min at 37 °C. For correlative light- and cryo-electron

microscopy, cells were chemically fixed in 2% PFA + 0.05% GA in 0.2 M Hepes for 15 min followed by fixation in 4% PFA in 0.2 M Hepes for 15 min and kept hydrated in PBS buffer prior to vitrification. For cell vitrification, fluorescently labeled CAD and SH-SY5Y cells were blotted from the back side of the grid for 7 s and rapidly frozen in liquid ethane[51] using a Leica EMGP system.

**EM immuno-labeling.** CAD cells were plated on grids as described in above. After incubation O/N at 37 °C, cells were treated with 25µM CK-666 for 15 min at 37 °C and fixed with PFA 4% for 15 min at 37 °C, quenched with 50 mM NH4Cl for 15 min, and blocked with PBS containing 1% BSA (w/v) for 30 min at 37 °C. Cells were labeled with a rabbit anti-N-Cadherin ABCAM 76057 antibody (1:200), followed by Protein A-gold conjugated to 10 nm colloidal gold particles (CMC, Utrecht, Netherlands). CAD cells were then rapidly frozen in liquid ethane[51] as above.

**Myo10 sample preparation.** CADs were transfected with GFP-Myo10 as described in the section Cell preparation and transfections and in Gousset et al.[27] In all, 250,000 cells were plated on grids, After O/N incubation at 37 °C, chemically fixed in 2% PFA + 0.05% GA in 0.2 M Hepes for 15 min followed by fixation in 4% PFA in 0.2 M Hepes for 15 min and labeled with WGA-546 (1:300 in PBS) for 15 min at RT; Before plunge-freezing, the grids were imaged with an inverted Zeiss Axiovision widefield microscope.

**Confocal microscopy and image analysis.** Fluorescent images from pharmacological assays were acquired using an inverted Zeiss LSM 700 confocal microscope. For quantification of TNT-connected cells, confluent CAD cells were mechanically detached and counted; 100,000 cells were plated for 24 h on Ibidi µ-dishes (Biovalley, France). Cells were then treated with CK-666 (25 µM or 50 µM, and DMSO-alone control treatments). Cells were treated for 15 min at 37 °C before fixation (15 min at 37 °C in 2% PFA, 0.05% glutaraldehyde and 0.2 M HEPES in PBS, and then additionally fixed for 15 min in 4% PFA and 0.2 M HEPES in PBS). Cells were carefully washed in PBS, labeled for 20 min at RT with a 3.3 µg/µL solution of Wheat-Germ Agglutinin (WGA) Alexa Fluor©-647 nm conjugate (Invitrogen) in PBS, washed again and sealed with Aqua-Poly/Mount (Polysciences, Inc.). The whole cellular volume was imaged by acquiring 0.5 µm Z-stacks with an inverted confocal microscope (Zeiss LSM 700) using ZEN software. TNT-connected cells, i.e., cells connected by straight WGA-labeled structures that do not touch the substrate and have a diameter smaller than 1 µm were manually counted by experimenters blind to the condition in the same manner as previously[19,23].

For vinculin-positive focal adhesions, after indirect immunofluorescence labeling of vinculin, described above in the section Immunofluorescence labeling, the bottom of the cell (in contact with the plastic dish) was imaged with an inverted confocal microscope (Zeiss LSM 700) using ZEN software. Displayed images correspond to stack projections Vinculin-positive peripheral focal adhesion were automatically detected and counted using ICY software (http://icy.bioimageanalysis.org/)[19,23].

**Scanning electron microscopy (SEM) and correlative light/SEM.** CAD and SH-SY5Y cells were plated overnight on gridded IBIDI dishes, fixed at 37 °C in 0.05% glutaraldehyde, 2% PFA in 0.2 M Hepes for 15 min followed by fixation in 4% PFA in 0.2 M Hepes for 15 min, stained with WGA 488 (1:300) for 15 min a RT. The samples were post-fixed in 2.5% glutaraldehyde in 0.2 M cacodylate buffer (pH 7.2) at 4 °C, washed three times 5 min in 0.2 M cacodylate buffer (pH 7.2), treated for 1 h with 1% osmium tetroxide in 0.2 M cacodylate buffer and then rinsed in distilled water. Samples were dehydrated through a graded series of 25, 50, 75 and 95% ethanol solutions for 5 min and for 10 min in 100% ethanol followed by critical point drying with $CO_2$. Samples were sputtered with a 10 nm gold/palladium layer and were observed in a JEOL JSM-6700F field emission scanning electron microscope at a voltage of 5 kV.

**Focused-ion beam scanning electron microscopy (FIB-SEM).** CAD and SH-SY5Y cells were plated on gridded Ibidi µ-dishes (Biovalley, France). Positions of TNTs were recorded by LM after labeling with WGA. Fixation was carried out as described for SEM followed by post-fixation with 1% (w/v) osmium tetroxide and 1.5% (w/v) potassium ferrocyanide for 30 min, incubation with 1% tannic acid for 30 min, followed by another post-fixation with 1% osmium tetroxide for 30 min. Samples were dehydrated in a graded ethanol series (25, 50, 75, 95 and 100%) and embedded in Epon (Agar Scientific, UK) and subsequently placed on a pin stub and covered with silver paint (Agar Scientific, UK). The surfaces were coated with a 10 nm-thick layer of gold/palladium using an ion Beam Coater (Gatan Inc, USA) to avoid charging effect. Three-dimensional tomography was performed with a FIB-SEM Auriga (Zeiss, Germany). An additional 1 µm protective layer of platinum was deposited on the surface of the region of interest using a 500 pA ion beam assisted deposition with a 30 kV acceleration potential. The cross-section was milled using a 10 nA ion beam current. The surface obtained was then polished using a 2 nA ion beam current. Tomographic sections of 10 nm in thickness during acquisition were obtained using a 500 pA ion beam current. SEM images were acquired with a voxel size of 10 x 10 x 10 nm for a frame size window of 2048 × 2048 pixels or with a voxel size of 2.5 x 2.5 x 10 nm with a frame size window of 4096 × 4096 pixels by

using a 1.5 keV acceleration voltage and a 30 µm aperture with an energy selected back-scattered electron detector in a Zeiss system. Alignment of the acquired stack of images was done using ImageJ. Segmentations and measurements were performed using Amira 6.4 (Thermo Fisher Scientific, USA).

**Cryo-correlative light and electron microscopy (i-CLEM).** Vitrified TEM grids containing fluorescently labeled CAD cells were imaged on an epifluorescent Axiovert 200 M inverted microscope (Zeiss) equipped with a cryo-correlative stage (Cryostage2, FEI), 10 × (NA 0.3), 40 × (NA 0.6) and 63 × (NA 0.75, working distance 1.7 mm) long working distance air objectives and with a 482/18 (Excitation), 520/28 (Emission, green) filter cube. The cryo-correlative stage (MPI Biochemistry, Martinsried[52]) was mounted on the light microscope motorized stage. Samples were transferred from liquid nitrogen storage only when the temperature of the microscope cryo-correlative stage was below –170 °C. Fluorescent and phase contrast digital images at cryogenic temperatures were recorded with an AxiocamMRm camera (Zeiss). Following cryo-fluorescence imaging, the vitrified samples on grids were stored in liquid nitrogen until they were used for cryo-TEM. In all images, the brightness and contrast were adjusted in order to highlight TNTs.

**Correlative light and cryo-electron microscopy (ii-CLEM).** CAD and SH-SY5Y cells on TEM finder grids were chemically fixed after fluorescent labeling as described in the cell culture section. The samples on TEM grids were then positioned in glass-bottom dishes (MatTek Corporation, Ashland, USA) in Hepes buffer. Fluorescence and corresponding transmitted light z-stack images of cells were obtained by using the same epifluorescence Axiovert 200 M inverted microscope (Zeiss) equipped with 482/18 (Excitation), 520/28 (Emission, green) and a 546/12 (Excitation), 575–640 nm (Emission, red) filter cubes and a 40 × (NA 1.3, WD 0,21 mm) oil objective. Following fluorescence imaging, the samples on grids were immediately vitrified and stored in liquid nitrogen until they were used for cryo-TEM.

**Cryo-electron microscopy.** Cryo-electron tomography was performed on a Tecnai 20 equipped with a field emission gun and operated at 200 kV (Thermo Fisher company). Images were recorded using either Explore 3D or SerialEM software on a 4k x 4k camera (Ultrascan from Gatan) and a Falcon II (FEI, Thermo Fisher) direct electron detector, with a 14 µm pixel size. Tilt series of TNTs were acquired covering either an angular range of – 52° to + 52° (Fig. 4a and Supplementary Movie 8), -51° to 51° (Fig. 2k and Supplementary Movie 5), –30° to + 48° (Fig. 3f and Supplementary Movie 6) and tilt range of −28° to + 48° (Fig. 3g and Supplementary Movie 7) or in specific cases due to sample physical constraints with a reduced angular range (Fig. 2f–h, and corresponding Supplementary Movies 1, 2, and 3), with 2–4 degrees increment. The defocuses used were −6, −8, and −10 µm. Tilt series of filopodial protrusions were acquired covering a tilt range of −65° to + 66° (Fig. 5b, e, h, with corresponding Supplementary Movies 9, 10, 11, and Fig. 5k–i with corresponding Supplementary Movie 12), with 2 degrees increment and defocus of −6 µm. Tilt series of TNTs with mitochondria were acquired covering a tilt range of −51° to +67° (Fig. 6 and Supplementary Movie 15). All tilt series were acquired at magnifications of 25,000x or 29,000 ×, binning 2, corresponding to a pixel size of 0.742 or 0,634 nm, respectively.

**Live-series microscopy.** Live time series images were acquired with a 60 × 1.4NA CSU oil immersion objective lens on an inverted Elipse Ti microscope system (Nikon Instruments, Melville, NY, USA). Laser illumination was provided by 488 and 561 nm. Pairs of images were captured in immediate succession with one of two cooled CCD cameras, which enabled time intervals between 20 and 30 s per z-stack. For live cell imaging, the 37 °C temperature was controlled with an Air Stream Stage Incubator, which also controlled humidity. Cells were incubated with 5% CO2 during image acquisition.

**Image analysis and visualization.** Tomographic tilt series were processed using version 4.9.2 of IMOD[53]. Projections were pre-processed by hot pixel removal and rough alignment by cross-correlation. Final alignments were done by using 10 nm fiducial gold particles coated with BSA (BSA Gold Tracer, EMS). The reconstructions were obtained by using a weighted back-projection algorithm[54]. For visualization purposes, the reconstructed volumes were processed by a Gaussian filter. Surface rendering was done with Amira 6.2 software package. The diameter of iTNTs and filopodia was quantified by measuring the diameter of individual structures along the tube every 300 nm. Quantitative measurements of vesicle diameter on TEM micrographs or of TNTs' and actin filaments' distances on 3D tomographic reconstructions were done by using the open source program Fiji. Distances between actin filaments were obtained in two ways: (1) by selecting sub-regions in sections from tomographic reconstructions containing only parallel actin filaments and by extracting peaks in the corresponding Fourier transform; (2) by performing plot profiles of the selected sub-regions and by measuring the peak-to-peak distances and the full width at half maximum.

**Statistical analysis.** The results of image analysis from pharmacological assays were transferred to Prism (GraphPad). For more than two groups statistical

significance was assessed by a one-way ANOVA with Tukey correction. Differences were considered significant at $*p < 0.05$, $**p < 0.005$, or $***p < 0.0005$. Quantifications were done blind. Quantitative data depicted as (± SEM) mean standard deviation.

**Reporting Summary**. Further information on experimental design is available in the Nature Research Reporting Summary linked to this Article.

## Data availability

The source data underlying Figs. 2a, 2b, 6f, 6g and Supplementary Figures 2i, 3 are provided as a Source Data file. A Reporting Summary for this Article is available as a Supplementary Information file. All other data that support the findings of this study are available from the corresponding author upon request.

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

## Acknowledgements

We thank Seng Zhu for image analysis technical expertize and Christel Brou for valuable manuscript comments. We also gratefully acknowledge Gerard Péhau-Arnaudet (Ultrapole, Institut Pasteur) for his help in setting up freezing conditions of CAD cells during early stages of the project, and Remi Blanc from Amira (FEI, Thermo Fischer Scientific) for the automated segmentation of actin in Fig. 6 using the XTracing—Filament detection module. We thank the equipment excellence CACSICE for providing the Falcon II direct electron detector. We acknowledge the Ultrastructural Bio-Imaging facility at Institut Pasteur, member of the national infrastructure France-Bio-Imaging (FBI) supported by the French National Research Agency (ANR-10-INBS-04) and IBISA. This work was supported by grants from the Agence Nationale de Recherche (JPND Neutargets: ANR-14-JPCD-0002-01 and ANR-16 CE160019-01 NEUROTUNN), and the Equipe FRM (Fondation Recherche Médicale) 2014 (DEQ20140329557) to C.Z. D.C.C was supported by the Pasteur—Paris University (PPU) International PhD Program. A.P. was supported by fellowships from France Parkinson and the Fondazione Ermenegildo Zegna.

## Author contributions

A.S. performed all correlative, cryo-correlative light, and electron microscopy experiments and quantification, wrote the results, and prepared the figures. D.C.C. prepared cells for TEM experiments, performed FM experiments and live imaging, helped with acquisition, prepared figures, and wrote the manuscript. A.P. performed CK-666 pharmacological experiments with D.C.C., set experimental conditions for SH-SY5Y cells, performed N-Cadherin and Myosin 10 experiments in the revised version, performed experiments and helped with the image rendering of TEM tomograms, prepared figures, and wrote results. K.G. and E.D. contributed equally; set-up and prepared CAD cell cultures for TEM and SEM and discussed earlier experiments. S.C.-D. performed FIB-SEM experiments and the corresponding data rendering, C.S. performed SEM experiments in CAD cells. C.Z. conceived the project, supervised all the work, and wrote the manuscript. A.S., D.C.C., A.P., J.K.-L., and C.Z. discussed the results. All authors commented on the manuscript.

## Additional information

**Competing interests:** The authors declare no competing interests.

