## [Peer Review File · Nature Communications]

Reviewers' comments:

Reviewer #1 (Remarks to the Author):

Comments of "Mapping of TNTs using Cryo-Electron Microscopy Reveals a Novel Structure"

The manuscript NCOMMS-17-18902-T by Dr Zurzolo and colleagues investigated ultra-structure of membrane nanotube (TNT) in neuronal CAD cells. By using cryo-EM combined with light microscopy, they revealed that TNTs are comprised with a bundle of 100 nm thin individual TNTs (iTNTs). The high resolution imaging also demonstrated the presence of vesicles along iTNTs and their localization with actin bundles. In addition, the organization of F-actin in iTNTs and filopodia was demonstrated by cryo-electron tomography. In this study, the authors applied a novel method and provided high quality images to demonstrate the fine structure of TNT and cytoskeleton inside. However, only morphological description can not provide sufficient information to interpret this delicate structure, as list in following points.

Main points:

1. One of the main discoveries of the study is that TNTs in CAD cells were comprised with thin iTNTs. However, previous studies on the ultrastructure of TNTs in T-cells and PC12 cells showed that they were single tubular structure (see Nat Cell Biol. 10, 211 and Science 303, 1007). Does this indicate that iTNT is specific for CAD cells? Since the authors only presented data using CAD cells, it is important to check the ultrastructure of TNTs in other cell types.
2. One of big debates of TNT study is if both ends of a TNT are opened or one end is closed. This is particularly important for the study of TNT-dependent vesicle transport: how vesicle across the membrane border if one end of TNT is closed. Ultrastructure study definitely has the chance to pry into this question. In the schematic diagram Fig. 6b, the authors draw both statuses: some iTNTs have two open ends, and some have only one open ends. Therefore, to support the conclusion, they should check complete iTNTs from one cell to the other cell, and focus on the membrane tip contact sites. Furthermore, the identification of adhesion protein between tips of iTNT and target cell plasma membrane will be very helpful for us to understand the formation of iTNT.
3. The electron tomography showed electron-dense connection between iTNTs (Fig. 3d), how the authors considered they were cellular structure but not artificial structure during sample preparation? A further investigation is necessary to identify the composition of these electron-dense structures, protein or lipid residue? This will definitely promote our understanding on the molecular

mechanism of formation of those iTNTs and the transport of vesicles inside. Combined with Point 2 and 3, how DiD-labeled vesicles transfer through TNT in CAD cells (Supplementary figure 3) should be addressed.

4. The morphological difference (length and diameter) between TNT and filopodia is well known by using conventional microscopy. However, based on the data presented in the paper, the organization of actin filaments and the average distance between actin did not show difference between TNTs and filopodia. This gives us an impression that TNT and filopodia are similar structures according to their f-actin organization. The current manuscript did not explain clearly the fundamental difference of TNT and filopodia mentioned in the abstract.

Minor points:

1. In Fig. 1b, the WGA-staining of TNT is not clear.
2. In Fig. 5m, a cross-section of filopodia was shown. It is useful to show a cross-section of iTNT for the comparison of TNT to filopodia.
3. Regarding vesicles inside and outside of iTNTs, the diameter of them is similar with exosome's diameter. Were these vesicles transferred via iTNTs from cells or just exosome bind or uptaked by iTNTs? Is it possible to detect exosome markers CD63 and CD81 in these vesicles?

Reviewer #2 (Remarks to the Author):

In this manuscript Sartori et al. studied TNT using cryo-EM and tomography in conjunction with fluorescent microscopy. The authors describe, in detail, the technical approaches that they have used but also present some structural characteristics of TNTs. While I believe that TNTs are interesting cellular structures and there are much yet to discover, some issues should be resolved before reconsidering of the manuscript:

Major point:

1. The current version fails in emphasizing the biological implication of the findings. The novel findings are very modest.

2. For the work presented here the authors used very elegant experimental procedures, however it does not include technological development in CLEM. The technical section within the main text should be minimized.

3. The diameter of actin filaments, which is reported in this work, is 5.3 nm and in Fig 4F it is 4.7nm. These findings indicate a major technical problem as actin filaments are much thicker.

Other points

1. In page 2, the authors indicated that integrins are transported in filopodia. Since integrins are membrane proteins, it implies that vesicles are being transported in filopodia. This is against what is stated at a later stage of the manuscript

2. The authors show vesicles inside TNTs and out of these structures. This indicates that these vesicles are not really transported but rather exported out of the cell. In fact I cannot imagine how these vesicles be transported having the density of actin shown in TNTs.

3. p.6 Thin connections between iTNT are reported in fig. 3c. Are these significant structures? If they are, please provide some statistics (frequency, how many times it is found along a TNT ?) and show a collage of several connectors.

4. Peak extensions, shown in Fig. 3d, are not unique structures of iTNT. In fact, these are most likely artefacts of the procedure (chemical fixation and/or blotting prior to vitrification). If the authors disagree, please provide evidence supported by needed control experiments.

5. Page 6, what do you mean by "long flexible bundles"? The length of F-actin within TNT was not quantified nor reported.

6. Figure 4 b,f. The two images should show identical dimensions, however, the dark density in fig. 4f has the same dimension of the spacing in fig. 4b (F-actin should be dark).

7. Page 7. What do the authors mean by "short parallel bundle"? Please provide the statistics of filament length. The filopodium shown in fig. 5h is not a typical 'healthy' filopodium and therefore does not represent these structures (see fig. 5b,d,f).

8. The authors conclude that the results shown here reinforced that TNTs play a role in transfer cargo. On which figures the authors base this claim? It seems that no direct indications are shown for the transport of cargo along TNT.

9. Some of the data shown here was acquired using a very limited tilt range (68°), the authors may want to replace figure 3d-f with more complete data set.

Point by point response to reviewers

Below we list verbatim the concerns from the reviewer followed by an indented response.

Reviewers' comments:

Reviewer #1 (Remarks to the Author):

Comments of "Mapping of TNTs using Cryo-Electron Microscopy Reveals a Novel Structure"

The manuscript NCOMMS-17-18902-T by Dr Zurzolo and colleagues investigated ultra-structure of membrane nanotube (TNT) in neuronal CAD cells. By using cyro-EM combined with light microscopy, they revealed that TNTs are comprised with a bundle of 100 nm thin individual TNTs (iTNTs). The high resolution imaging also demonstrated the presence of vesicles along iTNTs and their localization with actin bundles. In addition, the organization of F-actin in iTNTs and filopodia was demonstrated by cryo-electron tomography. In this study, the authors applied a novel method and provided high quality images to demonstrate the fine structure of TNT and cytoskeleton inside. However, only morphological description can not provide sufficient information to interpret this delicate structure, as list in following points.

Response: We thank the referee for his/her assessment, and for useful comments that have pushed us to the limits of our capacities to reply to his/her concerns. As indicated in the letter to the editor we have performed many additional experiments, established the pipeline of CLEM analysis in another cell line (from a different specie and origin) and set up another highly challenging technique, (FIB-SEM), to demonstrate that TNTs are open structures. We also present live imaging data to corroborate our EM data. While our manuscript does not solve all the details underlying these fascinating structures, it presents the first high quality and challenging report of this type, bringing many novel findings which will be used by the field to push research on TNTs forward.

Main points:

1. One of the main discoveries of the study is that TNTs in CAD cells were comprised with thin iTNTs. However, previous studies on the ultrastructure of TNTs in T-cells and PC12 cells showed that they were single tubular structure (see Nat Cell Biol. 10, 211 and Science 303, 1007). Dose this indicate that iTNT is specific for CAD cells? Since the authors only presented data using CAD cells, it is important to check the ultrastructure of TNTs in other cell types.

Response: Thank you for highlighting this important point and suggesting the use of another cell line to confirm the data presented in this manuscript. To test whether iTNTs are not structures found in mouse neuronal CAD cells only, we employed similar structural analysis on SH-SY5Y cells, a human neuroblastoma cell line previously used for the study of TNTs (see references below), and found that, like CAD cells, SH-SY5Y cells also connect via TNTs comprised of iTNTs that contain vesicular structures.

These results, which confirm previous observations made in CAD cells, have been described in detail in a new section of our manuscript titled "SH-SY5Y cells connect via continuous and closed-invaginating connections" (See line 305 – 331 and Figure 6A – 6L).

In addition, preliminary data shown below for this reviewer indicates that by cryo-EM analysis SW13 cells (also human but of epithelial origin), also form iTNTs, demonstrating their existence in non-neuronal cells. Given that the establishment of our pipeline for each cell type involves a long process, the recent evidence describing the patho/physiological role of TNTs in SH-SY5Y cells (Dieriks et al., Sci Rep 2017) [Reference 25 in manuscript], and our personal scientific interest in the role of TNTs in neurodegenerative diseases (Gousset et al., Nat Cell Biol 2009, Abounit et al., EMBO 2016) we decided to focus on SH-SY5Y cells as a second model over SW13 cells.

M. L. Vignais, A. Caicedo, J. M. Brondello, C. Jorgensen, Cell Connections by Tunneling Nanotubes: Effects of Mitochondrial Trafficking on Target Cell Metabolism, Homeostasis, and Response to Therapy. Stem cells international 2017, 6917941 (2017). [Reference 4 in manuscript]

F. Smith, J. Shuai, I. Parker, Active generation and propagation of Ca²⁺ signals within tunneling membrane nanotubes. Biophysical journal 100, L37-39 (2011). [Reference 24 in manuscript]

B. V. Dieriks, T. I. Park, C. Fourie, R. L. Faull, M. Dragunow, M. A. Curtis, alpha-synuclein transfer through tunneling nanotubes occurs in SH-SY5Y cells and primary brain pericytes from Parkinson's disease patients. Scientific reports 7, 42984 (2017). [Reference 25 in manuscript]

Although we agree that the ultrastructural characterization of TNTs has been tried by others before (including in the original manuscript by Rustom et al., Science 2004, [Reference 1 in manuscript], Sowinski et al., Nat Cell Biol 2008, [Reference 15 in manuscript] cited as example by this reviewer, and others cited below), no published study has dived into the structure of TNTs at the depth as we have in this manuscript. No statistical analysis or details on the structure were previously provided (e.g. membrane definition, inside-content, etc). Most importantly, no correlative analysis showing that the structures observed in fluorescence microscopy were the one described by EM was shown. We believe that this is a huge leap forward in the current state-of-the-art.

Of note, prior structural studies have indeed shown that TNTs are singular processes. This could be due to the following factors:

(1) TNTs previously shown might not be representative of all TNTs.

By performing SEM analysis of CAD and SH-SY5Y cells we learned that most TNTs break during sample preparation and only few, presumably the most robust, survive the dehydration and resin embedding steps involved in SEM sample preparation. To illustrate these observations, we have added our observations and representative images to our manuscript (line 79 – 84 and Figure S1). The singular TNT presented by Rustom et al., Science 2004 displays a crooked morphology that might be the result of stress during sample preparation. The TNTs shown by Sowinski et al., Nat Cell Biol 2008 are closed-ended processes that do not fit the stringent criteria of TNTs as open-ended structures. In order to overcome these limitations and study the ultrastructure of TNTs, including the most fragile, in the closest possible physiological condition, and without jeopardizing their native composition, we employed the cryo-EM strategies described in our manuscript (Figure 1). Furthermore, cryo-EM approaches present the advantage of allowing the study of actin organization within TNTs which cannot be resolved by conventional EM methods (see Figure S1C).

(2) Singular TNTs might be less abundant.

Our cryo-EM data suggests the existence of singular connections (see line 97, Figure S2A for CAD cells and line 267 – 270, Figure S2B for SH-SY5Y cells). Single connections were, however, thicker than iTNTs and rare compared to the iTNT configuration described in our manuscript.

We have modified our schematic in Figure 6 (now Figure 7H) to illustrate these findings and show that TNTs can either display single or multiple arrangements.

A. Kumar, J. H. Kim, P. Ranjan, M. G. Metcalfe, W. Cao, M. Mishina, S. Gangappa, Z. Guo, E. S. Boyden, S. Zaki, I. York, A. Garcia-Sastre, M. Shaw, S. Sambhara, Influenza virus exploits tunneling nanotubes for cell-to-cell spread. Scientific reports 7, 40360 (2017). [Reference 16 in manuscript]

J. Lu, X. Zheng, F. Li, Y. Yu, Z. Chen, Z. Liu, Z. Wang, H. Xu, W. Yang, Tunneling nanotubes promote intercellular mitochondria transfer followed by increased invasiveness in bladder cancer cells. Oncotarget 8, 15539-15552 (2017). [Reference 17 in manuscript]

G. Okafo, L. Prevedel, E. Eugenin, Tunneling nanotubes (TNT) mediate long-range gap junctional communication: Implications for HIV cell to cell spread. Scientific reports 7, 16660 (2017). [Reference 18 in manuscript]

2. One of big debates of TNT study is if both ends of a TNT are opened or one end is closed. This is particular important for the study of TNT-dependent vesicle transport: how vesicle across the membrane border if one end of TNT is closed. Ultrastructure study definitely has the chance to pry into this question. In the schematic diagram Fig. 6b, the authors draw both statuses: some iTNTs have two open ends, and some have only one open ends. Therefore, to support the conclusion, they should check complete iTNTs from one cell to the other cell and focus on the membrane tip contact sites.

Response: We agree that a standing debate in the field of TNTs is whether these connections are open- or close- ended at ‘contact sites’. Our attempts to observe these regions of interest by cryo-TEM/ET were not conclusive because these regions were too thick (>500nm) to be imaged by cryo-TEM. We have described these limitations in lines 309 – 313. We have also added a supplementary figure (Figure S7) to illustrate the resulting electron micrographs from the acquisition of contact sites using our fixation and imaging pipeline (Figure 1). In order to tackle this obstacle, we have employed focused ion beam SEM (FIB-SEM) on SH-SY5Y cells and found that cells can connect via open- and close-ended TNTs that seem to poach into the membrane of the opposing cell (Figure 7A – 7D, Supplementary Video 14 and 7E – 7G, Supplementary Video 15, respectively). Understanding whether these connections represent two different types of TNTs or pre-fusion events will require further investigation. Our results have been described in detail in a new section we have added to our manuscript, “SH-SY5Y cells connect via continuous and closed-invaginating connections” (line 305 – 331). The existence of open-ended connection was also corroborated by the observation of mitochondria being transported via TNTs under live, fluorescence microscopy. We have also added these results to our manuscript under a section titled “SH-SY5Y cells connect via TNTs capable of transporting mitochondria” (see line 259 – 303). Representative images and videos illustrating these results have also been added to our manuscript (see Figures 6G, S6A, and Supplementary Video 11 – 12).

Furthermore, the identification of adhesion protein between tips of iTNT and target cell plasma membrane will be very helpful for us to understand the formation of iTNT.

Response: Prior studies have shown that cell adhesion proteins (such as N-cadherins,) (Lokar et al., Protoplasma 2010) form when nanotubes are in proximity to the target cell membrane. However, we have previously attempted to test the role of N-cadherin in the formation of TNTs (specifically to see if it would allow fusion at the tip) and our data indicates that this cadherins do not reside in TNTs or at their tips (Gousset et al 2013, and unpublished data). Therefore, at the moment, the mechanism underlying TNT fusion, or the precise adhesion/fusogenic molecules (proteins and/or lipids) playing a role in TNT-to-membrane adhesion /fusion are not yet known. Studying the mechanism of fusion of TNTs is “*per se*” a very challenging project considering the lability and dynamicity of these structures and would require interdisciplinary approaches including biophysics/imaging, genetics, and cell biology (which we are currently trying to establish in our lab) in order to identify putative candidates. Once we have a list of potential candidates we will definitively attempt to correlate these proteins by EM, however this is still a long path.

3. The electron tomography showed electron-dense connection between iTNTs (Fig. 3d), how the authors considered they were cellular structure but not artificial structure during sample preparation? A further investigation is necessary to identify the composition of these electron-dense structures, protein or lipid residue? This will definitely promote our understanding on the molecular mechanism of formation of those iTNTs and the transport of vesicles inside.

Response: Thank you for highlighting this point. We agree that studying the composition of the electron-dense connections observed would be very interesting and indeed important to understand the formation of iTNTs and how a bundle of multiple iTNTs could be held together. However, these investigations would require putative candidates, a new entire study and better microscope settings.

Of note, we do not believe that these structures are artifacts of sample preparation as the reviewer seems to suggest as we also observe them in wild-type conditions, (thus ruling out chemical fixation as a plausible artifact effector, see panel below).

On the other hand, we cannot rule out the possibility that plunge-freezing would create these types of artifacts, although we are inclined to believe that this is not the case as this approach is the closest possible to native conditions. With the current microscope setup we have available at our disposal, we can only speculate that iTNTs require these thin structures to maintain mechanical stability (see discussion, line 366– 369).

We could remove these observations from the manuscript as they are not essential for our story however we feel that pointing to novel structural features (even if not fully characterized) is important to stimulate the field for further in-depth studies. Therefore, in the new version of our paper we have clearly addressed these limitations; we hope that the composition, frequency, and significance of these connections will be addressed in future studies.

Combined with Point 2 and 3, how DiD-labeled vesicles transfer through TNT in CAD cells (Supplementary figure 3) should be addressed.

Response: It is not clear to us whether this is a technical / methodological concern so we have divided the reply in two parts

Re methodology to study DiD-labeled vesicle transfer via TNTs:

The transfer of DiD-labeled vesicles through TNTs in CAD cells is an assay we often employ in combination with fluorescence microscopy to test the functionality of TNTs. This assay has been described in detail in previous published work (see references below) and is now well established in the field. We have modified our materials and methods section “Effect of CK-666 in the Transfer of DiD-labeled Vesicles in Co-Culture System” (line 465 – 477) and cited all appropriate references to clarify our results shown in Supplementary Figure 3 (now Supplementary Figure 4).

Re the molecular mechanism by which vesicles move through TNTs:

This is not a question that we wanted to address in this paper. It should be addressed, specifically, by live imaging and genetic experiments. Indeed, preliminary data by us and others have reported by fluorescence microscopy the presence of myosin motors inside

TNTs (Myo X: Gousset et al., J Cell Sci 2013 [Reference 27 in manuscript]; Myo Va: Rustom et al., Science 2004 [Reference 1 in manuscript] and Zhu et al., J Cell Sci 2005). Further experiments on these and other possible motors will be necessary.

On the other hand, our data shows, for the first time, the presence of vesicular structures and of structurally identified organelles (e.g. mitochondria, see below) inside TNTs. In our humble opinion, demonstrating what could not be shown by fluorescence microscopy (i.e. vesicular structures inside iTNTs) is a major advancement in the field. This piece of evidence strongly suggests that the transport of vesicles we previously described (see references below) takes place through the inside of the tube, and not the outside, or by “surfing” on its outer membrane. In addition, having also described in detail the arrangement of actin inside TNTs, these findings further corroborate our hypothesis that vesicles and organelles move on actin.

As previously stated, we cannot currently perform assays by cryo-EM using antibodies for myosins, or carry out experiments involving the overexpression of GFP-linked motors or knock-downs, as they would require another long set of experiments and further investigations that are clearly outside the scope of this paper.

Finally, in a new set of experiments, which now stand as a new section in our manuscript, “SH-SY5Y cells connect via TNTs capable of transporting mitochondria”, show that mitochondria can move through iTNTs. Representative images and videos describing these observations can be found in Figures 6G, S6A, and Supplementary Video 11 – 12. This is the first demonstration that mitochondria is inside TNTs strongly suggesting that it could move on actin, inside TNTs. In addition, these results support the observations we made of vesicular structures inside iTNTs and indicate that the transport of different cargo takes place inside TNTs and not at their surface.

How this occurs specifically is the next challenge!

E. Delage, D. C. Cervantes, E. Penard, C. Schmitt, S. Syan, A. Disanza, G. Scita, C. Zurzolo, Differential identity of Filopodia and Tunneling Nanotubes revealed by the opposite functions of actin regulatory complexes. Scientific reports 6, 39632 (2016). [Reference 19 in manuscript]

K. Gousset, E. Schiff, C. Langevin, Z. Marijanovic, A. Caputo, D. T. Browman, N. Chenouard, F. de Chaumont, A. Martino, J. Enninga, J. C. Olivo-Marin, D. Mannel, C. Zurzolo, Prions hijack tunnelling nanotubes for intercellular spread. Nature cell biology 11, 328-336 (2009). [Reference 22 in manuscript]

S. Abounit, E. Delage, C. Zurzolo, Identification and Characterization of Tunneling Nanotubes for Intercellular Trafficking. Current protocols in cell biology 67, 12 10 11-21 (2015). [Reference 23 in manuscript]

K. Gousset, L. Marzo, P. H. Commere, C. Zurzolo, Myo10 is a key regulator of TNT formation in neuronal cells. Journal of cell science 126, 4424-4435 (2013). [Reference 27 in manuscript]

4. The morphological difference (length and diameter) between TNT and filopodia is well known by using conventional microscopy. However, based on the data presented in the paper, the organization of actin filaments and the average distance between actin did not

show difference between TNTs and filopodia. This gives us an impression that TNT and filopodia are similar structures according to their f-actin organization. The current manuscript did not explain clearly the fundamental difference of TNT and filopodia mentioned in the abstract.

Response: The reviewer is correct in inviting caution in the interpretation of the results obtained regarding the f-actin organization of iTNTs vs. filopodia. We have changed the title of the section “Comparison of TNTs to Filopodia” to “Comparing and contrasting TNTs to filopodia” (line 218- 257) and carefully reworded our observations to underscore the differences observed.

On the other hand, we disagree that the morphological differences between TNTs and filopodia have been well characterized by conventional microscopy. In fact, there is little evidence describing differences between these two processes besides the fact that TNTs are generally longer compared to filopodia. Most of these observations were made by fluorescence and light microscopy studies and it is difficult to analyze the structural features of TNTs by these means. Therefore, we found this comment harsh and not acknowledging of the in-depth work we have performed here compared to published studies.

By employing cryo-ET we found that the actin diameter, width, and arrangement of filopodia and iTNTs were similar. However, as detailed in the reply to reviewer 2, on the scale of individual non-consecutive iTNTs tomograms (covering an area of $\sim 1.5 \times 1.5 \mu\text{m}^2$), we observed uninterrupted actin filaments organized in parallel bundles that stretched across the whole tomogram. This appears to be different in filopodia where, within the tomogram volume, actin filaments were organized in bundles comprised of shorter filaments with a length varying between 300 and 1100 nm, with only 15% of the filaments having a length larger than 1 μm (see diagram below in response to reviewer 2).

In addition filopodia also exhibited branched actin arrangements, which iTNTs did not, suggesting different regulatory complexes assembling their actin fibers. In support of this, our group previously demonstrated for the first time that actin regulatory complexes playing a role in the formation of filopodia actually had a different effect in the formation of TNTs, (Delage et al., Sci Reports 2016) [Reference 19 in manuscript].

Finally, a major difference between filopodia and TNTs is that filopodia are always single protrusions that do not contain membranous structures within, as observed for iTNTs (see lines 170 –173 and Figure 5).

Perhaps we failed to describe these observations if the reviewer cannot identify these major differences, which were never previously characterized or reported. We have also re-written this section and hope that our results are clearer this time.

Minor points:

1. In Fig. 1b, the WGA-staining of TNT is not clear.

Response: CAD cells connected by two TNTs shown in Figure 1b were frozen and fluorescently imaged under cryogenic conditions (i-CLEM) (See Figure 1a), an approach which yielded fluorescent images low on resolution compared to fixed samples imaged by confocal microscopy. We have now replaced Figure 1b with the corresponding lower magnification (10x) cryo-fluorescent image where the fluorescent signal of TNTs is clearly distinguishable. ii-CLEM on the other hand allowed higher spatial resolution fluorescence microscopy as a result of using oil immersion objectives with high numerical apertures (see TNT connecting CAD cells in Figure 3B and 3G).

2. In Fig. 5m, a cross-section of filopodia was shown. It is useful to show a cross-section of iTNT for the comparison of TNT to filopodia.

Response: We agree that a cross-section of iTNTs would help us better compare TNTs to filopodia. Therefore, we have added a panel to Figure 4 (Figure 4A2) which shows a cross section of two iTNTs displayed in Figure 4A1. The resolution obtained from the cross-section of iTNTs was not sufficient to extract information useful for comparison. This is presumably due to the lack of resolution achievable by our cryo-TEM and/or the considerable technical challenges we faced when imaging iTNT bundles under cryogenic conditions compared to isolated filopodia. We described these challenges in our discussion section (see line 392 – 425).

3. Regarding vesicles inside and outside of iTNTs, the diameter of them is similar with exosome's diameter. Were these vesicles transferred via iTNTs from cells or just exosome bind or uptaken by iTNTs? Is it possible to detect exosome markers CD63 and CD81 in these vesicles?

Response: Our fixation protocol for cryo-TEM assays, which is described in our materials and methods section “Cell preparation for cryo-EM” (lines 490 – 501), is suited to preserve TNTs and can only be combined with diffusible dyes (e.g. WGA). The use of antibodies in our experimental workflow is not feasible as it would involve a permeabilization step that would alter/destroy the ultra-structure of iTNTs. On the other hand, the overexpression of these proteins in cells could alter the wild-type phenotype and therefore not provide insights that we can compare to other observations made in this study.

Although this is an interesting question this was not the focus of our study. The presence of exosomal markers inside TNTs has been previously shown by others by FM (Hood et al., Lab Invest, 2009; Mineo et al., Angiogenesis, 2012; Taverna et al., Int J Cancer, 2012) and is therefore a very interesting question that should be further pursued in other in-depth studies.

Intriguingly, when we tested tetraspanin markers for exosomes, CD63 and CD81, in CAD and SH-SY5Y cells, but we observed an accumulation of these proteins at the ends of TNTs (see fig. below). This raises the question regarding the involvement of tetraspanins and exosomes in the fusion process of TNTs, which can also be supported by other studies in the literature (Thayanithya et al., Exp Cell Res, 2014). This question is highly interesting, and although we could add this FM data to our manuscript, we feel that it would require an in-depth analysis, which is out of the scope of this work. Therefore, we prefer to propose this as an intriguing hypothesis that we hope triggers the interest of the field in follow-up studies.

Phalloidin / CD81

Reviewer #2 (Remarks to the Author):

In this manuscript Sartori et al. studied TNT using cryo-EM and tomography in conjunction with fluorescent microscopy. The authors describe, in detail, the technical approaches that they have used but also present some structural characteristics of TNTs. While I believe that TNTs are interesting cellular structures and there are much yet to discover, some issues should be resolved before reconsidering of the manuscript:

Major point:

1. The current version fails in emphasizing the biological implication of the findings. The novel findings are very modest.

Response: Previous studies over the last decade published by our and other groups have shown that TNTs could play a role in several physiological and pathological processes due to their remarkable ability to transfer a wide variety of cargo (See Baker, Nature 2017 [Reference 12 in manuscript]). Some of these cargoes include organelles, pathogens, ions, genetic material, and misfolded proteins (see references 1 – 11). Their biological implication is significant and we have remodeled our introduction and discussion sections to accurately reflect the timely relevance of this work.

While this referee might suggest that our work does not have a biological implication or relevance, we disagree, especially considering the additional work that we have performed in this revised version. This body of work contributes to existing gaps in the field in several novel ways:

- 1) For the first time identified the structure observed by immunofluorescence in a correlative manner,
- 2) It demonstrates that TNTs are not singular units bridging two cells; instead, they are made up of smaller subunits, (iTNTs);
- 3) Identifies vesicular structures and mitochondria inside iTNTs, indicating that the cargo transfer described over the last decade (references above) take place inside TNTs, and not on the outside, or by ‘surfing’ on an outer membrane (see also reply point 3 to referee 1);
- 4) Identifies open-ended tubes, providing proof that TNTs are contiguous cytoplasmic connections that can yield cargo transfer;
- 5) Shows similarities, major, and minor differences between iTNTs and filopodia.

Together, these findings address important questions that have until now lacked in the field using challenging and complementary approaches (Cryo-CLEM, FIB-SEM and Live microscopy) for the study of these novel structures.

2. For the work presented here the authors used very elegant experimental procedures, however it does not includes technologic development in CLEM. The technical section within the main text should be minimized.

Response: Thank you. We have minimized non-essential technical details from our introduction and “Structural analysis of TNTs using cryo-correlated approaches” section (line 78 – 136). While the CLEM technique is not new, the way we used it here allows for the preservation and observations of structures never shown before.

3. The diameter of actin filaments, which is reported in this work, is 5.3 nm and in Fig 4F it is 4.7nm. These finding indicate a major technical problem as actin filaments are much thicker.

Response: In this work, we measured -by image analysis- the diameter of actin filaments both in iTNTs (5.3 ± 0.7 nm) and filopodia (5.7 ± 0.8 nm) of neuronal CAD cells. We obtained consistent values (within the experimental error) and within the published diameter range of actin filaments, previously shown to vary between 5 and 9 nm. For example, please see the cited reference below.

S. Aramaki, K. Mayanagi, M. Jin, K. Aoyama, T. Yasunaga, Filopodia formation by crosslinking of F-actin with fascin in two different binding manners. Cytoskeleton 73, 365-374 (2016) [Reference 32 in manuscript].

In Figure 4b and 4f, actin filaments correspond to the dark stripes, while the inter-filament distance corresponds to the light stripes. For both Figures (4b and 4f) we drew a schematic and layed it over the images corresponding to corresponding dashed areas showing the average: 1) actin diameter (5.3 ± 0.7 nm), inter-filament distance (4.7 ± 1.1 nm), and distance between the centers of the filaments (10 ± 0.8 nm).

However, as referee 2 pointed out, we wrongly placed the dashed selection area in Figure 4f so that the light stripes appear to be the actin filaments while the dark stripes represent inter-filaments distances. This fact generated confusion about the values reported for the actin average diameter. In the revised version of the figure we have fixed this problem by placing our selection on the correct area and by drawing in the schematic the actin filaments in a dark color and the inter-filaments distance in a light color (this answers also “minor points – 6”).

Other points

1. In page 2, the authors indicated that integrins are transported in filopodia. Since integrins are membrane proteins, it implies that vesicles are being transported in filopodia. This is against what is stated at a later stage of the manuscript

Response: We agree that the sentence “(...) while some studies have shown that filopodia engage in the trafficking of individual receptors and integrins via actin-based motor proteins such as Myosin X towards their tip” may create confusion in the readership, thus we have

decided to remove it from our manuscript. Indeed these studies involved mainly integrin transport along the membrane of the filopodia and not by vesicular transport. Please see our updated introduction.

2. The authors show vesicles inside TNTs and out of these structuree. This indicates that these vesicles are not really transported but rather exported out of the cell. In fact I cannot imagine how these vesicles be transported having the density of actin shown in TNTs.

Response: Our data indeed shows that iTNTs contain a high density of actin within. Vesicles observed inside iTNTs appear to ‘bulge’ out of the plasma membrane and push actin filaments to make their way through an iTNT (line 191 – 199, Figure 1, 2, and 3).

There is a bulk of evidence supporting the movement of vesicles and organelles via TNTs provided by fluorescence microscopy, live imaging, and other assays (see reply to point 3 from referee #1). Therefore, the goal of our CLEM analysis was to explore -for the first time- the structural features of this transport. The fact that we observed vesicular structures inside TNTs and beside actin filaments supports the role of TNTs in vesicular transfer. Further studies will need to explore how this event occurs in more detail.

Importantly, the vesicular structures we observed in our datasets were found inside iTNTs, suggesting that the transport of vesicles we previously described takes place through the inside of the tube, and not the outside, or by “surfing” on its outer membrane. Previous fluorescence-based studies have suggested the presence of myosin motors inside TNTs (Myo X: Goussset et al., J Cell Sci 2013 [Reference 27 in manuscript]; Myo Va: Rustom et al., Science 2004 [Reference 1 in manuscript] and Zhu et al., J Cell Sci 2005), which further corroborate our hypothesis that vesicles and organelles move on actin. We believe this should be specifically addressed in other studies and hope that our work will stimulate research in this direction.

We also verified that cargo, such as mitochondria, could not only travel through TNTs via live fluorescence imaging (see supplementary videos 11 – 12) but also transport inside iTNTs (see Figure 6, panels h-n). As with vesicles, mitochondria was not observed on the outside of iTNTs but instead in the inside. We have added a new section to our manuscript, ‘SH-SY5Y cells connect via TNTs capable of transporting mitochondria’ and representative images and videos to describe these observations in detail (see line 259 – 303, Figures 6G, S6A, and Supplementary Video 11 – 12).

3. p.6 Thin connections between iTNT are reported in fig. 3c. Are these significant structures? If they are, please provide some statistics (frequency, how many times it is found along a TNT ?) and show a collage of several connectors.

Response: As replied to reviewer 1 in point #3, these are qualitative observations that we could remove from the manuscript if the reviewers feel that they are too preliminary. Indeed, an in-depth analysis would require a novel study and other microscopic means. The difficulty of imaging multiple connectors along the entire length of a TNT is mainly due to technical drawbacks inherent to cryo-TEM described in lines 201 – 203. Firstly, while imaging at high resolution a region of interest (with a typical size of 1.2-1.5 μm), the adjacent area would in fact undergo severe beam damage; we could not acquire tomograms on consecutive areas. Secondly, not every position along a TNT meets the requirements for cryo-electron tomography: ice thickness < ~500 nm and the absence of obstacles coming into the field of view while tilting the specimen. Nonetheless, we have

multiple examples of these connections and therefore believe that they are an important structural feature of TNTs. For this referee, we have collected another example (see panels below; red arrowhead indicates thin connection).

From the data that we have acquired we can only suggest that these connectors do not appear to recur with a fixed periodicity. In terms of function, we can only speculate that iTNTs require these thin structures to maintain mechanical stability (see discussion, line 366 – 369).

Overall, we feel that pointing to novel structural features (even though not fully characterized) would be important to stimulate the field for further in-depth studies of TNT structure, formation, and mechanics. We hope that the composition, frequency, and significance of these connections will be addressed by us and/or others in future studies.

As previously mentioned in response to reviewer 1, point 3, we could remove these observations from the manuscript as they are not essential for our story however we feel that pointing to novel structural features (even if not fully characterized) is important to stimulate the field for further in-depth studies. We hope that the composition, frequency, and significance of these connections will be addressed in future studies.

4. Peak extensions, shown in Fig. 3d, are not unique structures of iTNT. In fact, these are most likely artefacts of the procedure (chemical fixation and/or blotting prior to vitrification). If the authors disagree, please prove provide evidences supported by needed control experiments.

Response: The observation of peak extensions connecting iTNTs in CAD cells was interesting but further experiments will be required to support their existence and role. We agree with this referee and in order to avoid any confusion in the readership regarding these processes we have decided to remove this panel from Figure 3. Please see our updated Figure 3.

5. Page 6, what do you mean by “long flexible bundles”? The length of F-actin within TNT was not quantified nor reported.

Response: As previously explained (in response to point #3), we were not able to image the complete actin bundle within the entire length of the TNT due to technical drawbacks inherent to cryo-TEM described in lines 201 – 203. That is, while imaging a region of

interest, the adjacent area would undergo severe beam damage. However, on the scale of individual non-consecutive iTNTs tomograms (covering an area of $\sim 1.5 \times 1.5 \text{ } \mu\text{m}^2$), we observed uninterrupted actin filaments organized in parallel bundles that stretched across the whole tomogram. This appears to be different in filopodia where, within the tomogram volume, actin filaments were organized in bundles comprised of shorter filaments, (see histogram below point 7) derived from automated template matching filaments recognition) with a length varying between 300 and 1100 nm, with only 15% of the filaments having a length larger than 1 μm .

We have also omitted the term 'flexible' as by EM we can in fact not measure actin filaments flexibility within bundles.

6. *Figure 4 b,f. The two images should show identical dimensions, however, the dark density in fig. 4f has the same dimension of the spacing in fig. 4b (F-ctin should be dark).*

Response: Thank you for pointing out this typo. We have replaced panel 4b and 4f with the appropriate panels. Please see the updated Figure 4.

In Figure 4b and 4f, actin filaments correspond to the dark stripes, while the inter-filament distance corresponds to the light stripes. For both Figures (4b and 4f) we drew a schematic and displayed it over the images corresponding to dashed areas showing the average: 1) actin diameter ($5.3 \pm 0.7 \text{ nm}$), inter-filament distance ($4.7 \pm 1.1 \text{ nm}$), and distance between the centers of the filaments ($10 \pm 0.8 \text{ nm}$). However, as this referee pointed out, we wrongly placed the dashed selection area in Figure 4f so that the light stripes appear to be the actin filaments while the dark stripes represent inter-filaments distances. This fact generated confusion about the values reported for the actin average diameter. In the revised version of the figure we have fixed this problem by placing our selection on the correct area and by drawing in the schematic the actin filaments in a dark color and the inter-filaments distance in a light color.

7. *Page 7. What do the authors mean by "short parallel bundle"? Please provide the statistics of filament length. The filopodium shown in fig. 5h is not a typical 'healthy' filopodium and therefore does not represent these structures (see fig 5b,d,f).*

Response: We agree that the expression "short filament bundle" is imprecise as we provide a -initial quantification of the distribution of the actin length within actin bundles in filopodia. Our results, (see histogram below), show shorter filaments in filopodia.

Concerning the filopodium shown in Figure 5G-K, we do not agree with the observation that the filopodium displayed is not typically “healthy” in comparison to what we show in Figure 5A-F. Filopodia in CAD cells have a length that spans between a few microns and 10-12 μm . In Figure 5A-F we compared the ends of filopodia; however, actin bundles could only be identified along the filopodium shaft. In this particular case, we acquired an image of the filopodium shaft located over a circular 2 μm hole carbon-support-film in order to increase the resolution of the tomogram. The somewhat distended appearance of the filopodium in the hole is in total agreement with what has been previously reported (see Aramaki et al., Cytoskeleton 2016) [Reference 32 in manuscript], where all quantifications were performed on filopodia segments laying on carbon holes and mostly with a distended appearance (see Aramaki et al, Figure 1b, Supplementary Figure 1b, and Supplementary Figure 2). Interfilament distance measurements were done on 7 bundles from distinct filopodia.

8. The authors conclude that the results shown here reinforced that TNTs play a role in transfer cargo. On which figures the authors base this claim? It seems that no direct indications are shown for the transport of cargo along TNT.

Response: Others and we have previously shown that TNTs are capable of transferring cargo (e.g. organelles, pathogens, ions, genetic material, and misfolded proteins) between cells (see line 34 – 37 and Ariazi et al., 2017, reference 5). By using fluorescence microscopy, FACS, and live imaging we have provided extensive evidence that CAD cells transfer vesicles, organelles, and amyloid proteins using TNTs. Other published reviews, including Abounit et al., 2012 [Reference 2 in manuscript] and Vignais et al., 2017 [Reference 4 in manuscript], provide extensive lists of different cell types forming TNTs and the various cargoes they transfer. However, the limit of resolution of fluorescence microscopy does not enable the magnification and resolution to determine whether the transfer occurs inside TNTs or on the limiting membrane. Thus, the findings presented in this manuscript, such as the observation of membranous compartments within iTNTs (Figure 1 – 3), supports with structural evidence the notion (acquired by live imaging) that

cargo can transfer through these thin structures.

As noted above, the vesicular structures we observed in our datasets were found inside iTNTs, suggesting that the transport of vesicles we previously described takes place through the inside of the tube, and not the outside, or by “surfing” on its outer membrane. We have modified the text to make this point more clear.

In addition, in the revised version of our manuscript, to confirm the functionality (e.g. transfer function) of TNTs bridging SH-SY5Y cells together we examined the transfer of mitochondria. While these cells were previously used by others, we never tested them and therefore used live imaging and found that these cells were also capable of transporting cargo through TNTs. These observations were further sustained by CLEM data showing, for the first time by tomography, mitochondria inside TNTs (see Figure 6, panels h-n). We have added a new section to our manuscript, ‘SH-SY5Y cells connect via TNTs capable of transporting mitochondria’ and representative images and videos to describe our new results in detail (Figures 6G, S6A, and Supplementary Video 11 – 12).

Finally, our FIB-SEM observations describing open-ended TNTs (Figure 7A – 7D, Supplementary Video 14) reinforce the notion that cargo transported via TNTs could reach the connected cell, as it appears from the movies, that although very suggestive do not provide the necessary resolution to see an open end.

We believe these additional results and previous published work provide sufficient evidence to suggest that the findings presented in this manuscript correspond to TNTs capable of transporting cargo (functional TNTs).

9. Some of the data shown here was acquired using a very limited tilt range (68°), the authors may want to replace figure 3d-f with more complete data set.

Response: We are aware that the data in Figure 3D and 3F were acquired with a limited tilt range due to geometrical constraints of the sample (i.e. the field of view was obscured when tilting at higher angles). Nevertheless, we believe the images shown in Figure 3 represent a unique example showing all the novel features that we highlight in iTNTs present in three non-consecutive areas of the same TNT.

We would like to point out that ALL of our other tomograms shown in the figures of our paper were acquired with a higher tilt range (90°-130°). In particular, image analysis was performed only on tomograms acquired with a high tilt range (110°-130°).

Reviewers' comments:

Reviewer #1 (Remarks to the Author):

The authors seriously responded to the comments and this reviewer's concern. In the revised manuscript, additional evidence was provided to show the presence of iTNTs and mitochondria transfer in SH-SY5Y cells. Overall, the main significance of this study is: 1. demonstrated that the membrane tubes in CAD and SH-SY5Y cells were comprised of a bundle of individual nanotubes. 2. revealed F-actin organization in iTNTs. However, some key questions still remain:

1. Although the authors showed the presence of iTNT in a second type of cells, I still want to know the difference between "classic" TNT and iTNT. In Davis paper (Nat Cell Biol. 10, 211), the diameter of TNTs in Jurkat cells were 180-380 nm. In the Science paper (Science 303, 1007), TNTs had a diameter of 50 to 200 nm. Both are in the same range as iTNT with a diameter of 200 nm. This gives the impression that iTNT is not a novel structures, but a regular single TNT as described previously. Then the connections between these neuronal cells were just a bundle of "classic" TNTs.

2. Many studies suggests there is membrane border at one end of TNT, which was proved by showing the presence of gap junctions on one end of TNTs. Here, the authors claimed that both ends of iTNT were open based on FIB-SEM observation. However, FIB-SEM can only scan specimen surface. How did it probe the membrane structure inside the tips of iTNTs? In addition, are the nanotubes shown in Fig7 D and G iTNTs or regular single TNTs?

3. The paper did not provide any molecular characteristic of iTNT besides actin. Therefore, it is hard to link the observation of iTNT with biological implication. The identification of protein candidates by immuno-gold staining will fill the gap between the morphological description and biological significance of the paper.

Reviewer #3 (Remarks to the Author):

In their manuscript entitled "Mapping of TNTs using Correlative Cryo-Electron Microscopy Reveals a Novel Structure" the authors describe observations of tunneling nanotubes using advanced microscopy methods. They describe the observed ultrastructure and compare it with those of filopodia.

The main findings of the presented work are:

- description of the detailed morphology of TNTs and their composition as a bundles of tubes.
- characterisation of intra-tubular vesicles.
- analysis of the actin organisation within the tubes.
- comparison with filopodia.
- discovery of open TNT connections between cells.

The manuscript describes well-considered and impressive application of challenging state-of-the-art technology and thereby provides novel scientific insights. While the cryo-correlative methods including tomography provide sound and convincing data, the FIB-SEM data that the authors provide is, in my eyes, not yet sufficient to prove the claim of open-ended connections between cells.

Recommendation:

Despite not being an expert in the field of TNTs and filopodia, I think the results of the presented work could have a significant impact. This impact, whose extent I cannot precisely judge, should be the key factor for the decision on acceptance of the manuscript.

The comments listed below describe necessary improvements from my point of view mostly concerning methods and comprehensibility of the article. Additional information to individual key results would in my eyes justify a publication also based on scientific novelty.

Major comments:

- writing:

The authors structure the results section of the manuscript like a progress report of the experimental procedure with a focus on techniques and justifying their application. This makes it hard for the reader to identify the ultimate scientific results. These are scattered throughout the manuscript with some results even appearing in the introduction (l.62 ff.). In contrast, the abstract, where I would expect a brief summary of the findings, remains very vague.

In my opinion it would be beneficial to first present the methods and their justification and then structure the remaining result section according to the scientific findings.

- key results:

the two observations that enthuse me for their novelty are the open connections between cells (see below) and the coiled filamentous threads that wrap the TNTs. These filaments seem to be directly exposed to the extracellular surrounding. Thus they are accessible to antibodies or other labeling strategies. As CLEM methods are employed, a fluorescence label might be sufficient. Another option would obviously be immuno-gold labeling for cryo-EM. I suggest that the authors test the “usual suspects” that these filaments could be composed of. Even a negative finding would significantly strengthen the observation and characterisation of these threads.

- FIB-SEM:

The quality of data provided in the supplementary movie 14 is insufficient to illustrate the claim of two connected open-ended tubes. The resolution of the data shown in SM 15 seems much better. It could simply be the presentation in the movie as a result of applied filters, contrast modification or compression of the file that leads to this. Another explanation might be problems in focus. I would strongly encourage the authors to provide high-resolution raw data of some key regions in the volume that illustrate the continuity of the plasma membrane. The engulfed tube (Suppl. Movie 15) would be ideal to demonstrate how the chosen acquisition parameters allow identification and tracing of membranes. Please also address these questions:

- Why was this cell type chosen for the FIB-SEM study?

- With good imaging quality 10nm target resolution should be sufficient to identify membrane connectivity/volume separation. Is the quality of the presented raw data sufficient to show the plasma membrane nicely?

- How many connecting tubes were observed in total? How many of these are open-ended?

Figures:

The amount of panels in the figures of the manuscript ranges from 8 to 25 (including zoom-in boxes, Fig. 3). With the additional coloured segmentations and up to 5 differently-coloured image annotations, this gets extremely confusing. With microscopy being the main tool for this research, I would certainly encourage the authors to remove the histograms from the main figures and focus more on the actual data.

In Figure 3, column D2/E2/F2 which does not provide much additional information could be removed or replaced with the individual high-magnification zoom panels.

Also, more concise figure captions will help the reader in understanding the findings.

Minor comments:

- provide significant figures: There are a number of quantitative measurements stated in the manuscript. Often, the amount of digits given with their S.D. is over-precise. Please use only the significant figures as defined by the SD. It would be also informative to know the total number of data points used in these measurements.

- Filament thickness: l. 94 specifies it for "thicker TNTs", what is it in regular ones?

- Actin organisation: l.203 ff. gives very detailed quantitative figures for the actin organisation in filopodia. For TNTs this characterisation is missing.

- l. 243 f.: I don't understand this sentence.

- l. 425: "min" is missing an i.

- l. 433 f.: "Positions were ... by fluorescence microscopy." Please specify. Is it the same approach as described above for confocal?

- l. 438: "recovered" should be replaced by "covered".

- l. 443: please include the beam current used for platinum deposition.

-l. 445: is this the correct beam value current for milling during acquisition?

- l. 447: please use the correct detector name e.g. "EsB", energy selected back scatter detector in a Zeiss system

- l. 452: "Vitrified TEMgrids..." Please specify the grid preparation procedure.

- l. 457 ff. "Briefly, ..." This sentence is unclear.

- Fig. 1 E/F/G/H, "Green Stars" should be arrows. Panels E+F are unnecessary. The authors could instead provide zoomed images of certain prominent ultrastructural features.

- Fig. 4 A1, A3 lack scale bars

- Fig. 5 B1/F1, Please provide a close-up view of the branching (red arrows).

Reviewer #4 (Remarks to the Author):

The manuscript by Dr. Zurolo (Sartori-Rupp et al.) on the ultra-structure of tunneling nanotubes (TNTs) has been significantly revised and improved in response to the previous critiques. This study uses the latest in advanced correlative light and electron microscopy techniques to determine the structure of novel and relatively uncharacterized thin intercellular bridges. The data demonstrates that while these structures appear by light microscopy to be a single tube many are in fact composed to collections of multiple smaller tubes (iTNTs) and vesicles and mitochondria are located with iTNTs and some vesicles are found between them. This study also contributes significant data that addresses a debate in the field of whether these connections are open or closed ended and how they relate to other actin structures, namely filopodia. The image analysis is very beautiful as well as detailed. While this study is a descriptive characterization of TNT structure very little is known about the actual structure of these delicate connections. Recent studies have indicated the importance of TNTs in human health and disease and much more needs to be understood about these structures. Therefore, based on the novelty of the discovery and the importance of detailed analysis of these structures to this new field makes this study worthy of publication in Nature Communication.

I recognize that this is a revised manuscript and I do not believe that additional experiments are required. However, I feel that several additions, outlined below, would greatly enhance the impact of this manuscript.

1. TNTs appear to be formed by different mechanisms in different cell types and this may have consequences on the TNT structure. Since the current analysis reported here has only been performed in detail on neuronal cell types I feel that the title should reflect this fact and a more detailed discussion on this should be included.

2. I would recommend that the different structural parameters between CAD and SH-SY5Y cells and the comparison to filopodia should be included as a table to easily high-light the similarities and differences to the reader.
3. Data is shown indicating that vesicles appear to be inside the iTNTs as well as between the iTNTs. A previous reviewer queried whether these vesicles were exosomes and I also wonder if they could be plasma membrane derived microvesicles. It is beyond the scope of this manuscript to include more data on this but a more detailed discussion should be included.
4. Furthermore, other studies have suggested that TNTs can be formed by twinning of extensions from both cells including PC12 cells (work of the Gerdes lab in 2009 FEBS Letters) and this twinning may result in the multiple iTNTs observed and should be discussed.
5. One further item would be to discuss the possible role of microtubules in iTNTs. Microtubule do not appear to be present, however microtubules are required for the transfer of vesicular material, originally identified in immune cells (Onfelt et al., J. Immunol. 2006).

We would like to thank the Reviewers' for their comments that have contributed to increase our novel findings and greatly raised the level of our paper. Below the point by point reply to their comments and the detailed changes in this re-revised version of the paper.

Point by Point reply:

Reviewer #1 (Remarks to the Author):

The authors seriously responded to the comments and this reviewer's concern. In the revised manuscript, additional evidence was provided to show the presence of iTNTs and mitochondria transfer in SH-SY5Y cells. Overall, the main significance of this study is: 1. demonstrated that the membrane tubes in CAD and SH-SY5Y cells were comprised of a bundle of individual nanotubes. 2. revealed F-actin organization in iTNTs. However, some key questions still remain:

1. Although the authors showed the presence of iTNT in a second type of cells, I still want to know the difference between "classic" TNT and iTNT. In Davis paper (Nat Cell Biol. 10, 211), the diameter of TNTs in Jurkat cells were 180-380 nm. In the Science paper (Science 303, 1007), TNTs had a diameter of 50 to 200 nm. Both are in the same range as iTNT with a diameter of 200 nm. This gives the impression that iTNT is not a novel structures, but a regular single TNT as described previously. Then the connections between these neuronal cells were just a bundle of "classic" TNTs.

We disagree with the reviewer in saying that iTNTs are not a novel structure and are just a bundle of classic TNTs, as iTNTs were never previously described.

iTNTs exhibit an architecture more complex than just a bundle of TNTs as iTNTs display structures that either link individual tubes or threads that coil around them, arguing for a specific and novel structure (see response to point #3).

This architecture also tells us something about the physiology of TNTs themselves (e.g. mechanism of formation and how TNTs work) as iTNTs might:

(I) play a role in making TNTs more robust and resistant to shearing forces.

(II) allow the bidirectional movement of vesicles (per TNT, but unidirectional in iTNTs).

(III) allow the formation of sequential iTNTs that run on top of each other once the first one forms in any given direction.

These hypotheses are discussed in our manuscript and are open avenues for further research that need to be explored (see lines 332-357).

We also argue that until the ultrastructure of the different protrusions found in different cell types described in the literature is resolved, we are not sure of what kind of connections we are looking at. Here we demonstrate, in two different neuronal cell lines, that what we previously described by fluorescent microscopy as TNTs, are comprised of iTNTs, which are open-ended, contain actin and vesicular cargoes, and are therefore novel structures.

The published work mentioned by the reviewer, also cited in our manuscript,

(I) only shows four, two for Science 303, 1007 and two for Nat Cell Biol. 10, 211, examples of TNT-like connections at the nano-scale resolution, with limited quantitative information. Given the small number of examples provided in the articles cited by the reviewer, it is plausible that the authors only "caught" single TNTs.

(II) These assays were performed with classic EM methods, which, as we show in our supplemental material, jeopardize the structure of TNTs by breaking fragile iTNTs and leaving behind only single thicker connections. It is possible that the EM methods they used tore down all iTNTs. After numerous futile attempts of testing conventional EM strategies, we found that cryo-TEM was best suited for the preservation of these fragile structures. It is worth pointing out that our report is the first one to implement a combination of several EM strategies to better characterize these biological structures.

(III) These studies were not done in correlative mode therefore we don't know what we are observing; that is, whether the EM structures shown corresponds to structures previously described using fluorescence microscopy.

Moreover, our work stands out by showing:

(I) The actin architecture within TNTs, never previously reported, and showing for the first time that actin assembles in a bundle of long parallel filaments, not branched and short as in filopodia. These findings also open up further investigations: What are the actin modifiers and bundlers that make up this actin structure and arrangement?

(II) Vesicular structures and mitochondria within TNTs at the nanoscale resolution. This was never shown in a correlative fashion, as the resolution of fluorescence microscopy cannot validate whether the vesicles and organelles are inside or on outside/top of TNTs.

(III) Mitochondria inside TNTs in the absence of microtubules. This implies that mitochondria inside TNTs move on actin. This is a new discovery and paves the way for further studies.

2. Many studies suggest there is membrane border at one end of TNT, which was proved by showing the presence of gap junctions on one end of TNTs. Here, the authors claimed that both ends of iTNT were open based on FIB-SEM observation. However, FIB-SEM can only scan specimen surface. How did it probe the membrane structure inside the tips of iTNTs?

We understand that this comment may be due to the fact that this reviewer is not familiar with the novel FIB-SEM technology. In addition to the 'specimen surface', FIB-SEM is also capable of imaging (probing) the inside of a cell by a method known as "slice & view", which boils down to two principles: (1) removal of a thin layer from a sample embedded in resin using a focused ion beam (FIB column) and (2) imaging the obtained surface with an electron beam, (SEM column). By repeating these steps, FIB-SEM provides 3D stacks of the region of interest, (the contact sites between cells and TNTs in our case), and their internal content (for review see Narayan et al., 2015).

Careful observation of the 3D volumes we obtained by FIB-SEM showed open-ended TNTs (see lines 291-305) (Figure 7 and supplementary Video 16-17).

However, and very interestingly, we found one close-ended protrusion's tip poking into the opposite cell (see lines 306-311) (Supplementary Fig. 7 and Video 18). This clearly demonstrates that by using this technique we can distinguish open- from close- ended tubes.

In addition, are the nanotubes shown in Fig7 D and G iTNTs or regular single TNTs?

Fig 7D (no longer in Figures) and 7G (now Supplementary Figure 7C) show single TNTs.

Figure 7 was updated with more representative examples obtained from acquisitions of both CAD and SH-SY5Y cells (see line 291-305) (Figure 7).

Our new example shown in Figure 7A-C displays three iTNTs connecting two CAD cells (see lines 291-300) (Supplementary Video 16).

Our new example shown in Figure 7D-F displays two iTNTs connecting two SH-SY5Y cells (see lines 301-305) (Supplementary Video 17).

Our old example shown in Supplementary Figure 7 displays two connections, one single open-ended TNT and one single close-ended protrusion (see lines 306-311) (Supplementary Video 18).

3. The paper did not provide any molecular characteristic of iTNT besides actin. Therefore, it is hard to link the observation of iTNT with biological implication.

The identification of protein candidates by immuno-gold staining will fill the gap between the morphological description and biological significance of the paper.

The finding that iTNTs contain vesicles and mitochondria on actin already held enormous biological implications and represented a large improvement from current published work; therefore, we disagree with the reviewer that our data did not provide any molecular characterization.

However, in order to further characterize iTNTs we have followed the suggestion of this reviewer and of reviewer 3 and performed immunogold labeling as suggested. Specifically, we tested a known transmembrane adhesion protein in neuronal cells, N-Cadherin, which was previously shown to be present in TNTs of other cells (Lokar et al., 2010,). By immunofluorescence we confirmed that N-Cadherin is found in TNTs connecting CAD cells. We then performed gold-immunolabelling on CAD cells against N-cadherin and localized gold nanoparticles on the connections between parallel iTNTs and at the base of filamentous threads coiled around iTNTs (see lines 155-160; 164-166) (Figure 3 and Supplementary Video 4). We believe that these results are very important for the understanding of the mechanics, formation and function of iTNTs within TNTs.

In addition, we have asked whether the unconventional motor protein MyoX, which was previously shown to increase the number of TNTs between CAD cells and vesicle transfer (Gousset et al., 2013), was localized inside iTNTs. Our observations demonstrate, for the first time, that vesicles inside iTNTs co-localize with GFP-MyoX puncta detected by fluorescence microscopy, supporting this motor plays a role in the formation of iTNTs and vesicle transfer. (see lines 177-183) (Figure 3D-H and Supplementary Video 6-7).

Please see detailed changes to the manuscript at the bottom of this page.

Reviewer #3 (Remarks to the Author):

In their manuscript entitled "Mapping of TNTs using Correlative Cryo-Electron Microscopy Reveals a Novel Structure" the authors describe observations of tunneling nanotubes using advanced microscopy methods. They describe the observed ultrastructure and compare it with those of filopodia.

The main findings of the presented work are:

- description of the detailed morphology of TNTs and their composition as a bundles of tubes.***
- characterisation of intra-tubular vesicles.***

- analysis of the actin organisation within the tubes.***
- comparison with filopodia.***
- discovery of open TNT connections between cells.***

The manuscript describes well-considered and impressive application of challenging state-of-the-art technology and thereby provides novel scientific insights. While the cryo-correlative methods including tomography provide sound and convincing data, the FIB-SEM data that the authors provide is, in my eyes, not yet sufficient to prove the claim of open-ended connections between cells.

We thank this referee for judging our work impressive and clearly listing all the novelties of our work. For the specific point on FIB-SEM, we have obtained new evidence using FIB-SEM that better illustrates the findings described in our manuscript. However, we have carefully replied in full to all his/her comments, which we believe have raised conspicuously the level of our manuscript, (see responses below to specific points).

Recommendation:

Despite not being an expert in the field of TNTs and filopodia, I think the results of the presented work could have a significant impact. This impact, whose extent I cannot precisely judge, should be the key factor for the decision on acceptance of the manuscript.

The comments listed below describe necessary improvements from my point of view mostly concerning methods and comprehensibility of the article. Additional information to individual key results would in my eyes justify a publication also based on scientific novelty.

Major comments:

-writing:

The authors structure the results section of the manuscript like a progress report of the experimental procedure with a focus on techniques and justifying their application. This makes it hard for the reader to identify the ultimate scientific results. These are scattered throughout the manuscript with some results even appearing in the introduction (l.62 ff.). In contrast, the abstract, where I would expect a brief summary of the findings, remains very vague.

In my opinion it would be beneficial to first present the methods and their justification and then structure the remaining result section according to the scientific findings.

We have condensed our introduction, modified our results, and restructured our discussion to improve the flow our manuscript; please see detailed changes to the manuscript at the bottom of this page.

- key results:

the two observations that enthuse me for their novelty are the open connections between cells (see below) and the coiled filamentous threads that wrap the TNTs. These filaments seem to be directly exposed to the extracellular surrounding. Thus they are accessible to antibodies or other labeling strategies. As CLEM methods are employed, a fluorescence label might be sufficient. Another option would obviously be immuno-gold labeling for cryo-EM. I suggest that the authors test the "usual suspects" that these filaments could be composed of. Even a negative finding would significantly strengthen the observation and characterisation of these threads.

We thank the reviewer for making these observations; however, we are unsure regarding what s/he means by "usual suspects" as TNT-specific markers do not exist at the moment. We presume that when referring to "testing the usual suspects" the reviewer is hinting at testing possible adhesion molecules. Thus, we tested a known transmembrane adhesion protein in neuronal cells, N-Cadherin, which was previously

shown to be present in TNTs of another cell type (Lokar et al., 2010). By immunofluorescence we confirmed that N-Cadherin is found in TNTs connecting CAD cells. We then performed gold-immunolabelling on CAD cells against N-cadherin and localized gold nanoparticles on the connections between parallel iTNTs and at the base of filamentous threads coiled around the iTNTs (see lines 155-160; 164-166) (Figure 3 and Supplementary Video 4).

We thank this referee for pushing us to do these experiments, (which took more time than we would have liked), but are, as we also believe, very important for the understanding of the mechanics, formation, and function of iTNTs within TNTs (see discussion lines 337-346).

While this discovery opens the way to more experiments to understand the role of N-Cadherin in TNT formation, (which we will definitely perform), we believe that in order to fully characterize the components of these connections and threads in more detail, we will first need to unveil the specific composition of TNTs by isolating TNTs and mass spectrometry analysis. We have already begun performing experiments in this sense (Proteomics. 2018 Jun;18(11):e1700294. doi: 10.1002/pmic.201700294. Epub 2018 Apr 16). Once these candidates become available, we plan to perform correlative structural studies using these specific targets. However, this project is well beyond the scope of this work and will take up a significant amount of time.

- FIB-SEM:

The quality of data provided in the supplementary movie 14 is insufficient to illustrate the claim of two connected open-ended tubes. The resolution of the data shown in SM 15 seems much better. It could simply be the presentation in the movie as a result of applied filters, contrast modification or compression of the file that leads to this. Another explanation might be problems in focus. I would strongly encourage the authors to provide high-resolution raw data of some key regions in the volume that illustrate the continuity of the plasma membrane. The engulfed tube (Suppl. Movie 15) would be ideal to demonstrate how the chosen acquisition parameters allow identification and tracing of membranes. Please also address these questions:

- Why was this cell type chosen for the FIB-SEM study?

SH-SY5Y cells were initially chosen as a different model of neuronal cells where TNTs had been observed by FM in order to sustain our observations in CAD cells. We performed the initial FIB-SEM experiments in these cells as they displayed more resistance to EM sample preparation than CAD cells. However, since our last submission, although it was not specifically requested, we have produced FIB-SEM volumes of open-ended TNTs connecting CAD cells (see Figure 7, panels A-C; Supplementary Video 16), which confirm our observations in SH-SY5Y cells.

- With good imaging quality 10nm target resolution should be sufficient to identify membrane connectivity/volume separation. Is the quality of the presented raw data sufficient to show the plasma membrane nicely?

The 10nm/pixel resolution was chosen to compromise resolution and speed of acquisition. It was sufficient to observe membrane connectivity.

We have acquired our CAD example (Supplementary Movie 16) at 2.5nm/px resolution in xy to better illustrate the continuity of the plasma membrane (see line 291-300).

We have replaced movie 14 in our supplementary material with another movie (Supplementary Movie 17) which better illustrates TNT-to-cell membrane contact sites (see line 301-305). We have also shown the contact sites in Figure 7C and 7F. Finally, we have uploaded the raw data as a single .tiff file for both SH-SY5Y and CAD FIB-SEM examples, which can be accessed via the following link:
<https://drive.google.com/open?id=1xAuMc4IP09swRMt1najGERubhWheHpnq>

- How many connecting tubes were observed in total? How many of these are open-ended?

Please refer to the table below for the number of cells and open- and close-ended tubes observed by FIB-SEM.

	SH-SY5Y	CAD
iTNTs	1	8
Single TNTs (open)	7	2
Single TNTs (engulfed tube)	1	0
Single TNTs (closed attach to the cell surface)	1	1
Single TNTs (closed without contact with the other cell)	2	2

Figures:

The amount of panels in the figures of the manuscript ranges from 8 to 25 (including zoom-in boxes, Fig. 3). With the additional coloured segmentations and up to 5 differently-coloured image annotations, this gets extremely confusing. With microscopy being the main tool for this research, I would certainly encourage the authors to remove the histograms from the main figures and focus more on the actual data.

The quantifications of our observations provide meaning to our results; therefore, we disagree with this comment and believe that they should remain in the manuscript.

In Figure 3, column D2/E2/F2 which does not provide much additional information could be removed or replaced with the individual high-magnification zoom panels.

We hope to provide the readers of our work with enough resources to be able to understand our findings. Panels D2/E2/F2 of Figure 3 (now Figure 2, panels F2, G2, H2, respectively) illustrate the 3D depth of our tomograms, which is only displayed in these panels. We therefore believe these panels and insets should remain as part of this figure. We clarified this in our figure legend.

Also, more concise figure captions will help the reader in understanding the findings.

We have updated and condensed figure captions to help the reader in understanding our findings.

Minor comments:

- provide significant figures: There are a number of quantitative measurements stated in the manuscript. Often, the amount of digits given with their S.D. is over-precise. Please use only the significant figures as defined by the SD.

We thank the reviewer for this suggestion. We have applied changes to our quantitative measurements according to this suggestion.

It would be also informative to know the total number of data points used in these measurements.

The total number of data points is detailed in the chart below:

	CAD	SH-SY5Y
Vesicle size	80	68
iTNTs, average diameter	468	457
iTNTs, distance between iTNT surfaces	202	N/A
iTNTs, distance between the center of actin filaments	44	N/A
iTNTs, distance between the center of actin filament (cross-section)	74	N/A
iTNTs, distance between actin surfaces (cross-section)	76	N/A
Filopodia, distance between actin filaments	44	N/A

- Filament thickness: l. 94 specifies it for “thicker TNTs”, what is it in regular ones? Individual TNTs are 123nm thick (SD=66nm).

These quantifications are described in detail in our results, section “N-Cadherin decorates threads holding together iTNTs in a bundle”, (see lines 142-150).

- Actin organisation: l.203 ff. gives very detailed quantitative figures for the actin organisation in filopodia. For TNTs this characterisation is missing.

Under the imaging conditions used, quantitative information such as the number of filaments per bundle and actin bundling proteins could not be clearly detected for TNTs. This is possibly due to the lack of resolution achievable by our cryo-TEM on thick samples and the considerable technical challenges we faced when imaging iTNT bundles under cryogenic conditions compared to isolated filopodia.

Since our last submission however, we made use of a new camera, which allowed us to look at cross-sections of iTNTs. From new tomograms of TNT-connecting CAD cells (Figure 4 and Supplementary Video 4), we have learned that the distance between the centers of adjacent actin filaments and distance between their surfaces was 10 and 5 nm, respectively (see lines 198-200). We have also ensured that the terminology used to describe these two measurements (i.e. distance between the center of adjacent actin filaments and distance between their surfaces) is consistent across our manuscript and published literature (Aramaki et al., 2016).

- l. 243 f.: I don't understand this sentence.

This sentence has been carefully reworded to: “Intriguingly, mitochondria were only observed in one of the iTNTs. As with vesicles, mitochondria also created a bulge in the iTNT containing it” (see line 277-278).

- I. 425: "min" is missing an i.

Done

- I. 433 f.: "Positions were ... by fluorescence microscopy." Please specify. Is it the same approach as described above for confocal?

Correlative cryo-EM experiments were all performed using fluorescence microscopy with the exception of the example displaying mitochondria in TNTs connecting SH-SY5Y cells (see details below). We have rephrased our Materials and Methods section "Focused-ion Beam Scanning Electron Microscopy (FIB-SEM)" to clarify the procedure used (see lines 520 - 539).

Fluorescence and confocal microscopy are not used "in lieu" of one another throughout our manuscript, figures, and supplementary material. Figures 1B, 2D, 2I, 3D, 7, and Supplementary figures 1B1, 1F1, 5E1, 7A-B were obtained using epi-fluorescence microscopy whereas Figure 6A, 6H-J and Supplementary figure 1A, 3A, 3C, 3E, 4A, 5A, were acquired by confocal microscopy. Figure legends accurately mention where each microscopy method was used.

- I. 438: "recovered" should be replaced by "covered".

The word we initially intended to use was, recorded, not recovered. Change applied (see line 522).

- I. 443: please include the beam current used for platinum deposition.

The beam current used for platinum deposition was: 500pA assisted deposition with a 30 kV acceleration potential (see line 527-528).

- I. 445: is this the correct beam value current for milling during acquisition?

The correct value for milling is 500pA, not 500 μ A. We have applied this change to our manuscript (see line 530).

- I. 447: please use the correct detector name e.g. "EsB", energy selected back scatter detector in a Zeiss system

This has already been corrected in our manuscript (see lines 533-534).

- I. 452: "Vitrified TEMgrids..." Please specify the grid preparation procedure.

We describe this procedure in detail in our Materials and Methods section 'Cell preparation for Cryo-EM' (see lines 473-483).

- I. 457 ff. "Briefly, ..." This sentence is unclear.

This sentence has been carefully reworded to: "The cryo-correlative stage (MPI Biochemistry, Martinsried) was mounted on the light microscope motorized stage. Samples were transferred from liquid nitrogen storage only when the temperature of the microscope cryo-correlative stage was below -170 °C. " (see lines 543-546).

- Fig. 1 E/F/G/H, "Green Stars" should be arrows. Panels E+F are unnecessary. The authors

could instead provide zoomed images of certain prominent ultrastructural features.

Green stars are now arrows.

We wanted to provide the readership with images that helped with the visualization of our data by outlining iTNTs (Figure 1E-F), while also indicating the numerous observations we made inside iTNTs (Figure 1G-H). Merging outlines with arrows/stars clogged our images. Therefore, we prefer to display these 2 images separately.

- Fig. 4 A1, A3 lack scale bars

Figure 4 has been modified since our last submission. Figure 4A has been removed from our figures given that it did not provide much additional information to our actin analysis of iTNTs (lines 190-197).

- Fig. 5 B1/F1, Please provide a close-up view of the branching (red arrows).

We have added insets with close-up views of the regions mentioned to our figure. Please see our updated Figure 5 B1/F1.

Reviewer #4 (Remarks to the Author):

The manuscript by Dr. Zurolo (Sartori-Rupp et al.) on the ultra-structure of tunneling nanotubes (TNTs) has been significantly revised and improved in response to the previous critiques. This study uses the latest in advanced correlative light and electron microscopy techniques to determine the structure of novel and relatively uncharacterized thin intercellular bridges. The data demonstrates that while these structures appear by light microscopy to be a single tube many are in fact composed to collections of multiple smaller tubes (iTNTs) and vesicles and mitochondria are located with iTNTs and some vesicles are found between them. This study also contributes significant data that addresses a debate in the field of whether these connections are open or closed ended and how they relate to other actin structures, namely filopodia. The image analysis is very beautiful as well as detailed. While this study is a descriptive characterization of TNT structure very little is known about the actual structure of these delicate connections. Recent studies have indicated the importance of TNTs in human health and disease and much more needs to be understood about these structures. Therefore, based on the novelty of the discovery and the importance of detailed analysis of these structures to this new field makes this study worthy of publication in Nature Communication.

I recognize that this is a revised manuscript and I do not believe that additional experiments are required. However, I feel that several additions, outlined below, would greatly enhance the impact of this manuscript.

1. TNTs appear to be formed by different mechanisms in different cell types and this may have consequences on the TNT structure. Since the current analysis reported here has only been performed in detail on neuronal cell types I feel that the title should reflect this fact and a more detailed discussion on this should be included.

Thank you for the suggestion. We have adjusted our title to better fit our work. Our new title is now: "Mapping of TNTs in Neuronal Cells using Correlative Cryo-Electron Microscopy Reveals a Novel Structure".

2. I would recommend that the different structural parameters between CAD and SH-SY5Y cells and the comparison to filopodia should be included as a table to easily high-light the similarities and differences to the reader.

We appreciate this comment but we realized that a table would be difficult to compile. Combined with the new data we have added to this revised version, we do not have the space for a new figure. However, we believe that this suggestion is very valuable and we are thinking of compiling such a table, which would include other results present in the literature, in a review that we plan to write.

3. Data is shown indicating that vesicles appear to be inside the iTNTs as well as between the iTNTs. A previous reviewer queried whether these vesicles were exosomes and I also wonder if they could be plasma membrane derived microvesicles. It is beyond the scope of this manuscript to include more data on this but a more detailed discussion should be included.

We thank the reviewer for these suggestions. We have added additional sentences to our discussion but given that we have not explored this possibility and the limited space we have, we were not able to discuss this at length, (see lines 368-373).

4. Furthermore, other studies have suggested that TNTs can be formed by twinning of extensions from both cells including PC12 cells (work of the Gerdes lab in 2009 FEBS Letters) and this twinning may result in the multiple iTNTs observed and should be discussed.

We thank the referee for this suggestion; however, previous studies, including ours, have observed that both paired cells can form TNTs. These never suggested that TNTs could form from multiple iTNTs coming together to form TNTs. On the other hand, we have described and discussed the evidence demonstrating this notion in discussion of our manuscript (see lines 338-340; - 349-352).

5. One further item would be to discuss the possible role of microtubules in iTNTs. Microtubule do not appear to be present, however microtubules are required for the transfer of vesicular material, originally identified in immune cells (Onfelt et al., J. Immunol. 2006).

In this revised version we suggest that TNTs observed in immune cells and containing microtubules might be different from the ones we observe here between neuronal cells. Therefore, ultrastructural analysis of these other structures is needed to assess this possibility (see lines 362-364).

Indeed, the TNTs that we observe in both neuronal cell lines seem to contain only actin. We have made this concept clearer in our manuscript. We found vesicles on top of actin bundles and seemingly connected to actin bundles by structures that could be myosin motors (Figure 2D-2H). To sustain this hypothesis and better characterize iTNTs, we have asked ourselves whether the unconventional motor protein MyoX, which was previously shown to increase the number of TNTs between CAD cells and vesicle transfer (Gousset et al., 2013), was localized inside iTNTs. Our observations demonstrate, for the first time, that vesicles inside iTNTs co-localize with GFP-MyoX puncta detected by fluorescence microscopy supporting a role for this motor in their formation and vesicle transfer. (see lines 175-183) (Figure 3D-H and Supplementary Video 6-7).

Finally, we also show that iTNTs between SH-SY5Y cells, in which we observed mitochondria (Figure 7 and Supplementary Video 13-14), always contain actin and do not contain microtubules. As we state in our manuscript (line 374-377) this represent a new discovery and supports some of the literature suggesting that mitochondria can travel on actin filaments (Morris et al., 1995; Quintero et al., 2009; Pathak et al., 2010) and (line 379).

Please find below the detailed changes in this re-revised version.

Detailed changes:

Figures:

Figure 2: "Imaging iTNTs in 3D Using Correlated Light and Cryo-Electron Tomography" corresponds to our previous Figure 3.

- Panel **A**: is the previously panel **J** of the figure 3.
- Panel **B**: corresponds to the graphic of distribution of iTNT-iTNT distance, previously shown in the Supplementary figure 5 "Spacing between iTNTs Ranges in Size".
- Panels **C-K2**: correspond to the previous panel in Figure 3A-I2.
- Panel **H1**: scale bar is added.

Figure 3: This is a new Figure: "Ultrastructural Analysis of N-cadherin and Myo10 in iTNTs."

Addition of the following new panels:

- Panel **A**: Low magnification of electron micrograph displaying N-Cadherin immunogold labeling of CAD cells connected by iTNTs.
- Panel **B1-B2**: High magnification cryo-tomography slices of panel **A**, attached Movie 4 (new movie).
- Panel **C**: Rendering of Tomogram in **B**
- Panel **D**: Epifluorescence micrographs of CAD cells overexpressing GFP-Myo10
- Panel **E**: Low magnification of electron micrograph corresponding the TNTs in **D**
- Panel **F-G**: High magnification cryo-tomography slices corresponding to the yellow dashed square in panel **E**; attached new Movie 6 showing tomogram of panel **F**, new Movie 7 showing tomogram panel **3G**.
- Panel **H**: Rendering of tomogram in panel **G**

Figure 4 "Cryo-Electron Tomography and Image Analysis Reveal F-actin Organization in iTNTs."

- Panel **A1**: Replaced with panel **A3**.
- Panel **A2**: Removed

Addition of the following new panels:

- Panel **G**: High magnification cryo-tomography slices of Figure **3B**
- Panel **H**: cross section through the cryo electron tomogram in panel **G**

Figure 7. old figure 7 "FIB-SEM of TNTs in SH-SY5Y Cells Reveals Open-Ended Contact Zones" is removed;

A new figure 7 has been made, "FIB-SEM of TNTs in CAD and SH-SY5Y cells open-open Contact sites

Addition of the following new panels:

- Panel **A, D**: Confocal micrograph of TNTs connected CAD cells(**A**) and SH-SY5Y cells (**B**) (left) overlay of confocal micrograph with FIB-SEM segmented rendition (right).
- Panel **B, E**: Fib-SEM volume rendering of **A** and **D**.
- Panel **C, F**: TNT-to-cell contact sites at the both end of connection in CAD and SHSY-5Y cells. Attached new Movie 16 for CAD FIB-SEM (panel **B-C**) and new Movie 17 for SHSY-5Y FIB-SEM (panel **E-F**).
- Panel **G**: Schematic diagram showing how to cells connect via TNTs; light gray semicircles were removed.

Supplemental Figures

- Supplementary Figure 1: "Morphology of TNTs in CAD and SH-SY5Y cells using fluorescence and scanning electron microscopy" is replaced by new Supplementary figure: "Morphology of TNTs in CAD Cells using Fluorescence, Scanning-, and Cryo- Electron Microscopy",

the following panels are removed and combined in the new Supplementary Figure 5:

- Panel **B**: Confocal micrographs of SH-SY5Y cells connected by TNTs (now panel **A** in Supplementary Figure 5).
- Panel **G**: Low and intermediate magnification scanning electron micrographs of SH-SY5Y cells connected by a thick TNT (now panel **B** in Supplementary Figure 5).
- Panel **H**: Low magnification scanning electron micrograph of SH-SY5Y cells showing broken protrusions (now panel **C** in Supplementary Figure 5).
- Panel **I**: Low magnification scanning electron micrograph of SH-SY5Y cells showing thin TNTs and broken protrusions (now panel **D** in Supplementary Figure 5).

New panel are combined in Supplementary 1:

- Panel **B**: previously panel C "confocal and transmitted light micrographs of CAD cells connected by TNTs."
- Panel **C** previously panel D "low and high magnification scanning electron micrographs of CAD cells shown in 'b'."
- Panel **D** previously panel E "low-magnification scanning electron micrograph of CAD cells connected by a thick TNT."
- Panel **E** previously panel F "low and high magnification scanning electron micrographs of CAD cells showing broken protrusions."
- Panel **F** previously panel **A** from the Supplementary Figure 2 "Morphology of Thick TNTs in CAD and SH-SY5Y Cells"
- Panel **G-H** previously panel **A-B** from the Supplementary Figure 3 "Cryo and Chemical Fixation of TNTs Yield Similar Structural Features."

Previously Supplementary figure 2 and 3 are removed and their panel combined in the new Supplementary Figure 1 and 5.

-Supplementary Figure 2: previously Figure 2 "Gallery of Different Membrane Compartments Observed Outside and Within iTNTs."

- Supplementary Figure 4: Addition of the new Figure 4 "N-cadherin detection on TNTs using CLEM."

New panel added:

- Panel **A**: Confocal micrographs displaying CAD cells connected by TNTs labeled with an anti-N-cadherin antibody
- Panel **B**: Low magnification electron micrograph of TNT-connected CAD cells immunogold labeled with an anti-N-cadherin.
- Panel **C**: High magnification cryo-tomography slice corresponding to the white dashed rectangle in B.

Deletion of Supplementary Figure 5 "Spacing between iTNTs Ranges in Size" this panel is now in the new Figure 2 panel **B**.

- Supplementary Figure 6: "Mitochondria travels through TNTs" is now Supplementary Figure 5 "Structural and Functional Analysis of TNTs in SH-SY5Y Cells using Fluorescence, Scanning and Cryo- Electron Microscopy, and Live Imaging."

Panel added:

- Panel **A**: Representative confocal micrographs displaying SH-SY5Y cells stained with WGA previously panel **B** in the old Supplementary 1
- Panel **B-D**: SEM micrograph of SH-SY5Y are the previously panel **G, H, I** of the old Supplementary Figure 1 "Morphology of TNTs in CAD and SH-SY5Y Cells using Fluorescence and Scanning Electron Microscopy".
- Panel **E**: Single, thick TNTs connecting SHSY-5Y cells is the previous panel **B** of the old Supplementary Figure 2. "Morphology of Thick TNTs in CAD and SH-SY5Y Cells."
- Panel **F**: displaying iTNTs connecting SH-SY5Y cells fixed by cryo fixation (rapid freezing) and chemical is the previously panel **C-D** of previously supplementary figure 3 "Cryo and Chemical Fixation of TNTs Yield Similar Structural Features."

- Panel **H**: Time-lapse images of wild-type SH-SY5Y cells stained with WGA (green) and MitoTracker (red) show mitochondria traveling across a TNT previously panel **A** of old Supplementary Figure 6.
- Panel **I, J, K**: are the previously panel **B, C, E** of previously Supplementary Figure 6.
- Panel **L, M**: are the previously panel **H, M** of previously Supplementary Figure 6.
- Panel **D** of previously Supplementary Figure 6 is removed.
- Previously Supplementary Figure 7: “Ends of TNTs” is now the new Supplementary 6 iTNT Contact Sites by cryo EM
- Supplementary Figure 7 is now “FIB-SEM of TNTs in SH-SY5Y Cells Reveal Open-Ended Contact Sites”
- panel **A**: Confocal micrograph of SH-SY5Y cells stained with WGA (green).
- panel **B**: Overlay of 3D rendering of FIB-SEM tomogram segmentations over fluorescent counterpart of panel **A**.
- Panel **C**: 3D rendering of FIB-SEM tomogram segmentation shown in panel **B**.
- The panel **A, B, C** are respectively the panel **E, F, G** of the previously Figure 7.

Supplemental Movies

- Movie 1 - Now corresponding to figure 2**F**
- Movie 2 - Now corresponding to figure 2**G**
- Movie 3- Now corresponding to figure 2**H**
- Movie 4- This is a new movie showing slices of a reconstructed tomogram displaying iTNTs connecting two CAD cells immunogold labeled with an anti-N-Cadherin that correspond to the Figure 3**B-C**
- Movie 5 - Now corresponding to figure Figure 2**K**
- Movie 6- This is a new movie showing slices of a reconstructed tomogram displaying iTNTs connecting CAD cells transfected with GFP-Myo10, correspond to the Figure 3**F**
- Movie 7- This is a new movie showing slices of a reconstructed tomogram displaying iTNTs connecting CAD cells transfected with GFP-Myo10, correspond to the Figure 3**G**.
- Movie 8- Now corresponding to figure Fig 4**A**
- Movie 9- Now corresponding to figure Fig 5**B**
- Movie 10- Now corresponding to figure Fig 5**D**
- Movie 11- Now corresponding to figure Fig 5**F**
- Movie 12 - Now corresponding to figure - Fig 5**H**
- Movie 13 - Now corresponding to figure - Fig 6**G**
- Movie 14 - Now corresponding to figure - Fig 6**H**
- Movie 15 - Now corresponding to figure - Fig 6**M**
- Movie 16 This is a new movie showing FIB-SEM tomographic volume of open-ended TNTs connecting two CAD cells, surface rendition, and manually annotated segmentation of the tomographic data created with the Amira software package, corresponding to figure 7**B-C**
- Movie 17 – This is a new movie showing FIB-SEM tomographic volume of open-ended TNTs connecting two SH-SY5Y cells, surface rendition, and manually annotated segmentation of the tomographic data created with the Amira software package, correspond Figure 7**E-F**
- Movie 18 - Now corresponding to Supplementary figure 7

Text (major only)

Abstract

Introduction

- Section reworded to highlight biological implications.

Results

- Section 2: “Quantitative ultrastructural analysis of TNTs using cryo-electron tomography” is been organized in a new section named “N-Cadherin decorates threads holding together iTNTs in a bundle” in which are described the additional findings.

- Section 5: "SH-SY5Y cells connect via continuous and closed-invaginating connections" is being remodeled in a new section: "Open-ended iTNTs connect the cytoplasm of cells" in we described the new data of FIB-SEM

Discussion

- Remodeling of section to highlight additional findings and how they fit with our previous observations and the existing literature. The overall conclusion, perspectives, and major impact have also been carefully revised.

REVIEWERS' COMMENTS:

Reviewer #3 (Remarks to the Author):

The authors of the submitted manuscript "Mapping of TNTs in Neuronal Cells using Correlative Cryo-Electron Microscopy Reveals a Novel Structure" have significantly re-structured the manuscript based on comments and suggestions by the reviewers. In its current format and structure it has become a lot more comprehensive and organized.

Based on the impressive scientific findings, I support publication given with the following major modification.

The quality and resolution of the presented FIB-SEM data, even though it has been extended and improved, does not allow conclusions about individual iTNTs. This fact however does not weaken the results and conclusions of the manuscript significantly. In the raw data that has been made available and also in Fig. 7C,F, connectivity of the membrane can be seen at several positions. Topologically this implies open-ended connections of the individual entities, hence the iTNTs. Still, this is not directly visible in the data. I strongly urge the authors to rephrase the affected sections of the manuscript to clarify this and make their data interpretation more credible. Currently the text does not clearly state whether the FIB-SEM observations correspond to iTNTs or a TNT in general, this makes it unnecessarily vague.

This would include the major conclusions (l. 72) where in my eyes "iTNTs" should be changed to "TNTs".

There is a strong risk that too high claims and over-interpretation of the limited FIB-SEM data will lead to a loss of credibility for the entire, otherwise very impressive, study.

Minor comments:

Fig 7 G: I suggest this panel to be a separate figure. It is summarizing the entire scientific story and does not only relate to FIB-SEM. Also, the labeling in the legend is wrong (F instead of G). Please also include the observation that iTNTs possibly merge/divide in the scheme.

Methods, FIB-SEM: l. 538 ff. Please specify the exact voxel sized in xyz. Are these volumes recorded with isotropic resolution?

Figure 3 G: what are the orange arrows pointing at? Please include them in the legend.

Figure S5: The legend is very confusing. Panels are sometimes referred to in the beginning of a sentence but also sometimes in the end.

Reviewer #4 (Remarks to the Author):

Recent studies have indicated the importance of TNTs in human health and disease. However, very little is known about how they are generated and very few studies have examined the structure of these thin versatile connections. This study contributes significant data that addresses a debate in the field of whether these connections are open or closed ended and how they relate to other actin structures, namely filopodia. The image analysis is very detailed and elegant and contributes novel information about these structures in neuronal cells. It will admit that this study is a descriptive characterization of TNT structure yet very little is known about the actual structure of these delicate connections. Therefore, I think that this study is a significant contribution to the field that is only strengthened in this revised version of the manuscript.

REVIEWERS' COMMENTS:

Reviewer #3 (Remarks to the Author):

The authors of the submitted manuscript "Mapping of TNTs in Neuronal Cells using Correlative Cryo-Electron Microscopy Reveals a Novel Structure" have significantly re-structured the manuscript based on comments and suggestions by the reviewers. In its current format and structure it has become a lot more comprehensive and organized.

Based on the impressive scientific findings, I support publication given with the following major modification.

The quality and resolution of the presented FIB-SEM data, even though it has been extended and improved, does not allow conclusions about individual iTNTs. This fact however does not weaken the results and conclusions of the manuscript significantly. In the raw data that has been made available and also in Fig. 7C,F, connectivity of the membrane can be seen at several positions. Topologically this implies open-ended connections of the individual entities, hence the iTNTs. Still, this is not directly visible in the data.

I strongly urge the authors to rephrase the affected sections of the manuscript to clarify this and make their data interpretation more credible. Currently the text does not clearly state whether the FIB-SEM observations correspond to iTNTs or a TNT in general, this makes it unnecessarily vague.

We thank the reviewer for suggesting us to be more clear and precise concerning the conclusions drawn from our FIB-SEM data on iTNTs. We have updated the corresponding section of our manuscript following this reviewer's suggestions, stating that the FIB-SEM observations correspond to TNTs in general.

This would include the major conclusions (l. 72) where in my eyes "iTNTs" should be changed to "TNTs".

We do agree with the reviewer and therefore replaced "iTNTs" with "TNTs".

There is a strong risk that too high claims and over-interpretation of the limited FIB-SEM data will lead to a loss of credibility for the entire, otherwise very impressive, study.

Minor comments:

Fig 7 G: I suggest this panel to be a separate figure. It is summarizing the entire scientific story and does not only relate to FIB-SEM.

Thank you. We agree and therefore have created a new figure for this panel.

Also, the labeling in the legend is wrong (F instead of G).

This panel no longer exists as it was turned into another figure, Figure 8.

Please also include the observation that iTNTs possibly merge/divide in the scheme.

We thank the reviewer for this suggestion. We have made adjustments to our schematic and updated the figure legend.

Methods, FIB-SEM: l. 538 ff. Please specify the exact voxel sized in xyz. Are these volumes recorded with isotropic resolution?

SEM images were acquired with a voxel size of 10nm x 10nm x 10nm for a frame size window of 2048 x 2048 pixels or with a voxel size of 2.5nm x 2.5nm x 10nm with a frame size window of 4096 x 4096 pixels.

Figure 3 G: what are the orange arrows pointing at? Please include them in the legend.

Orange arrows indicate vesicle-iTNT connections. We have updated the figure legend.

Figure S5: The legend is very confusing. Panels are sometimes referred to in the beginning of a sentence but also sometimes in the end.

We have made slight changes to this figure legend; however, we believe the flow of the legend accurately describes the figure.

Reviewer #4 (Remarks to the Author):

Recent studies have indicated the importance of TNTs in human health and disease. However, very little is known about how they are generated and very few studies have examined the structure of these thin versatile connections. This study contributes significant data that addresses a debate in the field of whether these connections are open or closed ended and how they relate to other actin structures, namely filopodia. The image analysis is very detailed and elegant and contributes novel information about these structures in neuronal cells. It will admit that this study is a descriptive characterization of TNT structure yet very little is known about the actual structure of these delicate connections. Therefore, I think that this study is a significant contribution to the field that is only strengthened in this revised version of the manuscript.

We thank the reviewer for highlighting the relevance of TNTs in biology and underscoring our findings.